# Semantic Tube Prediction: Beating LLM Data Efficiency with JEPA

Hai Huang[1]   Yann LeCun[2]   Randall Balestriero[3]

## Abstract

Large Language Models (LLMs) obey consistent scaling laws—empirical power-law fits that predict how loss decreases with compute, data, and parameters. While predictive, these laws are descriptive rather than prescriptive: they characterize typical training, not optimal training. Surprisingly few works have successfully challenged the data-efficiency bounds implied by these laws—which is our primary focus. To that end, we introduce the Geodesic Hypothesis, positing that token sequences trace geodesics on a smooth semantic manifold and are therefore locally linear. Building on this principle, we propose a novel Semantic Tube Prediction (STP) task, a JEPA-style regularizer that confines hidden-state trajectories to a tubular neighborhood of the geodesic. STP generalizes JEPA to language without requiring explicit multi-view augmentations. We show this constraint improves signal-to-noise ratio, and consequently preserves diversity by preventing trajectory collisions during inference. Empirically, STP allows LLMs to match baseline accuracy with $16\times$ less training data, directly violating the data term of fine-tuning scaling laws and demonstrating that principled geometric priors can surpass brute-force scaling. Code: https://github.com/galilai-group/llm-jepa#stp

## 1. Introduction

We argue that empirical scaling laws characterize *typical* rather than *optimal* training, suggesting the rigid power-law barrier is an artifact of current objectives. The core limitation is next-token prediction: a local objective that conflates surface statistical noise with global semantic signal. We propose a fundamental shift: explicitly constraining hidden

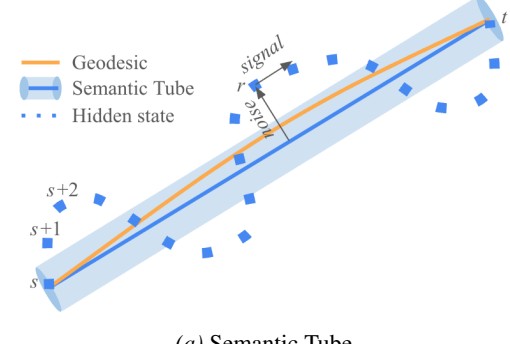

*(a) Semantic Tube*

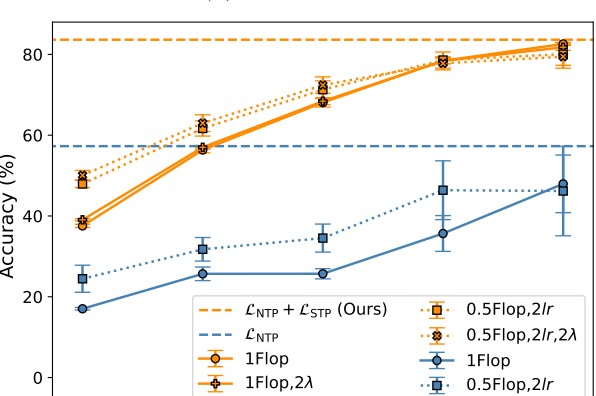

*(b) Data Efficiency*

*Figure 1.* Semantic Tube improves data efficiency. **(a)** We hypothesize that error-free hidden state trajectories are geodesics, which are locally linear and approximated by the Semantic Tube. The dotted line depicts a trajectory distorted by training loss. Deviations perpendicular to the tube constitute *noise*, while the component along the geodesic represents the *signal*. **(b)** With our approach ($\mathcal{L}_{\text{NTP}} + \mathcal{L}_{\text{STP}}$), accuracy shows a negligible drop when the training dataset is halved, and it matches full-dataset standard fine-tuning ($\mathcal{L}_{\text{NTP}}$) accuracy using only $\frac{1}{16}$ of the training data. In contrast, $\mathcal{L}_{\text{NTP}}$ degrades significantly when the dataset is halved.

state dynamics to separate the error-free semantic trajectory from this noise.

First, we formally demonstrate that, although tokens are discrete, token sequences can be modeled by an Ordinary Differential Equation (ODE). The Picard-Lindelöf (Existence and Uniqueness) Theorem (Coddington & Levinson,

[1]Palona AI [2]NYU [3]Brown. Correspondence to: Hai Huang <hai@palona.ai>, Randall Balestriero <randall_balestriero@brown.edu>.

*Proceedings of the 43rd International Conference on Machine Learning*, Seoul, South Korea. PMLR 306, 2026. Copyright 2026 by the author(s).

1955) guarantees that if the velocity is smooth enough, there is only one possible path forward from any starting point. In other words, trajectories originating from distinct initial states will never intersect. In the context of LLMs, if the ODE model holds, this implies that *error-free* generations from distinct prompts maintain their semantic separation, theoretically ruling out mode collapse and preserving diversity.

Next, we hypothesize that the Principle of Least Action (Lanczos, 1966) is at work. This principle states that the path taken by a system between two points minimizes the "Action" (the integral of the Lagrangian over time), resulting in a "straight line" or geodesic on the underlying manifold. We further hypothesize that, as the manifold is an artifact of the training process, it admits a smooth structure. Consequently, the geodesics are locally linear almost everywhere. In the context of LLMs, this implies that the trajectories of error-free token sequences—and by extension, the trajectories of error-free hidden states—are confined within a tube centered along a straight line.

We designate this structure the **Semantic Tube** (Figure 1) and leverage it to regularize the LLM training process. The Semantic Tube posits that the noise—which causes deviations from the error-free trajectories—concentrates along the directions perpendicular to the tube. Let $s < r < t$ denote the indices of three tokens. We define the *noise* term as $(h_r - h_s)_{\perp h_t - h_s}$, representing the component of $h_r - h_s$ perpendicular to $h_t - h_s$, and the *signal* term as $(h_r - h_s)_{\parallel h_t - h_s}$, representing the component parallel to $h_t - h_s$. Minimizing the noise term is expected to improve the Signal-to-Noise Ratio (SNR) during training. We formulate this as an auxiliary loss term, the Semantic Tube Prediction (STP) loss $\mathcal{L}_{\text{STP}}$, which can be seamlessly integrated into the training objective:

$$\mathcal{L} = \mathcal{L}_{\text{NTP}} + \lambda \cdot \mathcal{L}_{\text{STP}}$$

where $\mathcal{L}_{\text{NTP}}$ is the cross-entropy loss for Next Token Prediction (NTP) and $\lambda$ is a hyperparameter controlling the strength of the STP loss.

Semantic Tube draws inspiration from the Joint-Embedding Predictive Architecture (JEPA) (Assran et al., 2023; Baevski et al., 2022), which learns to predict the representation of one view based on another. In our approach, we postulate that any segment of a token sequence aligns with the global trajectory; consequently, the predictor reduces to an identity function.

If the Geodesic Hypothesis holds, it entails the following predictions:

- (P1) $\mathcal{L}_{\text{NTP}}$ alone is insufficient for high-quality generation. Consequently, we expect to observe $\mathcal{L}_{\text{NTP}}$ plateau even as $\mathcal{L}_{\text{STP}}$ continues to decrease.

- (P2) Semantic Tube improves SNR, resulting in superior data efficiency (Figure 1) and accuracy.

- (P3) Semantic Tube preserves diversity.

- (P4) We expect to see $\lambda \ll 1$ to accommodate instances where the geodesic deviates from a straight line.

- (P5) The identity function serves as a superior predictor compared to learned projections.

We conducted extensive experiments validating predictions (P1) through (P5). These results provide a strong indication that the Geodesic Hypothesis represents a simplified form of self-consistency for autoregressive sequence models. Furthermore, they confirm the validity of the noise/signal decomposition (Figure 1) and establish Semantic Tube as an effective self-supervised learning objective for LLMs.

## 2. Training and Inference Dynamics

In this section, we formally analyze training and inference dynamics, proposing that token sequence trajectories can be modeled by an Ordinary Differential Equation (ODE) characterized by ballistic trajectories.

### 2.1. Training ODE

Let $x_{\leq t}$ denote a token sequence of length $t$, where $x_t$ represents the $t$-th token, $h_t$ is the corresponding hidden state, and $f(\cdot)$ denotes the neural network such that $h_t = f(x_{\leq t})$. Each hidden state $h_t$ is subsequently unembedded to predict the next token $x_{t+1}$.

During training, the predicted token $u(h_t)$ may diverge from the ground truth $x_{t+1}$; this discrepancy constitutes the training loss. However, due to teacher forcing, we invariably feed the ground truth sequence $x_{\leq t+1}$ into $f(\cdot)$ to generate $h_{t+1}$. Consequently, assuming a converged network where the loss is minimized, the training dynamics can be modeled as:

$$x_{t+1} = \mathring{u} \circ \mathring{f}(x_{\leq t}) \qquad (1)$$
$$h_t = \mathring{f}(x_{\leq t}) + \epsilon_t \qquad (2)$$

where $\mathring{f}$ and $\mathring{u}$ represent the functions of the converged network, and $\epsilon_t$ denotes the residual unembedding error.

If a time-indexed variable $z_t$ follows the difference equation $z_{t+1} - z_t = g(z_t, t)$, it can be approximated by an ODE of the form $dz_t = g(z_t, t)dt$. While the hidden state dynamics in Equation (2) do not fit this form (as $h_{t+1}$ depends on the entire history $x_{\leq t}$ rather than just $h_t$), the sequence dynamics in Equation (1) do. Specifically, $x_{\leq t+1} = x_{\leq t} \oplus x_{t+1} = x_{\leq t} \oplus \mathring{u} \circ \mathring{f}(x_{\leq t})$, where $\oplus$ denotes concatenation. Letting $\ominus$ denote the prefix-removal operator, we obtain:

$$x_{\leq t+1} \ominus x_{\leq t} = \mathring{u} \circ \mathring{f}(x_{\leq t}).$$

This formulation closely resembles the update rule $z_{t+1} - z_t = g(z_t, t)$, suggesting that an ODE is a plausible model for the dynamics.

Although tokens are discrete, their embeddings lie in a continuous vector space $x_t \in \mathbb{R}^{d_{\text{model}}}$. Let $T$ denote the maximum sequence length; then the sequence resides in $\mathbb{R}^{T \times d_{\text{model}}}$. In Section A, we demonstrate that under specific arrangements, the operation $x_{\leq t+1} \ominus x_{\leq t}$ can be treated as vector subtraction $x_{\leq t+1} - x_{\leq t}$. This leads to the following proposition:

**Proposition 1** (Training ODE). *The LLM training process can be modeled as a solution in the token sequence space* $\mathbb{R}^{T \times d_{\text{model}}}$ *to the ODE:*

$$dx_{\leq t} = \mathring{u} \circ \mathring{f}(x_{\leq t})dt.$$

Proposition 1 models $x_{\leq t}$ as following a ballistic trajectory in $\mathbb{R}^{T \times d_{\text{model}}}$. The Picard-Lindelöf Theorem guarantees that if $\mathring{u} \circ \mathring{f}(\cdot)$ and its partial derivatives with respect to $x_{\leq t}$ are continuous, the ODE admits a unique solution for a given initial condition. Consequently, within this ODE framework, sequences generated from distinct prompts (initial conditions) cannot intersect, theoretically ruling out mode collapse, and preserving diversity.

### 2.2. Mode Collapse at Inference Time

Let $h^*$ denote the optimal trajectory of hidden states, defined as:

$$h_t^* = h_t - \epsilon_t = \mathring{f}(x_{\leq t}) \tag{3}$$

If $\mathring{f}(\cdot)$ is Lipschitz-continuous (Khalil, 2002), then the trajectory $h^*$ is also ballistic.

However, $\mathcal{L}_{\text{NTP}}$ alone may not suffice to drive $\epsilon_t$ to zero. Recall that the goal of $\mathcal{L}_{\text{NTP}}$ is to converge $u(h_t)$ to $x_{t+1}$. Since the hidden state $h_t$ is continuous while the token $x_{t+1}$ is discrete, the training process can be modeled as finding the correct Voronoi cell (Okabe et al., 2000), without stipulating the exact location within the cell. This flexibility is necessary for the Picard-Lindelöf Theorem to apply: as illustrated in Figure 2, it allows error-free geodesics ($h_t^*$) to traverse the same Voronoi cell at distinct locations, thereby avoiding intersection. Nevertheless, $h_t$ may drift onto an incorrect geodesic within the cell, leading to mode collapse.

This analysis indicates that $\mathcal{L}_{\text{NTP}}$ alone is insufficient for generation quality, strongly motivating an additional loss term ($\mathcal{L}_{\text{STP}}$) to explicitly minimize $\epsilon_t$. It also implies that within the correct Voronoi cell, $\mathcal{L}_{\text{NTP}}$ may plateau while $\mathcal{L}_{\text{STP}}$ continuously decreases. Therefore, (P1).

In Section B, we demonstrate that in the infinite-width limit (Yang & Littwin, 2021), the inference process can

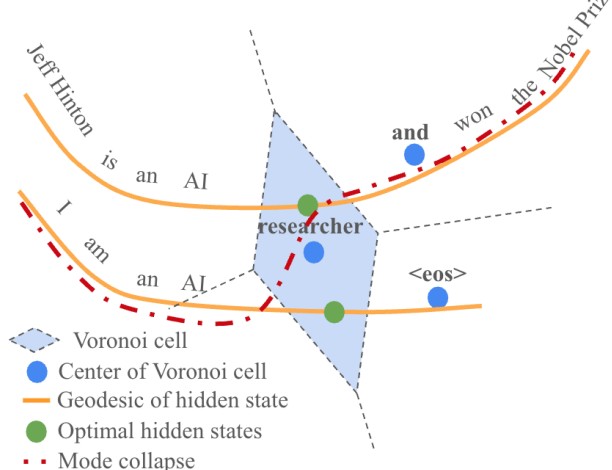

*Figure 2.* Two hidden state trajectories with similar prefixes pass through the Voronoi cell of the "**researcher**" token at different locations, leading to different next hidden states and hence different next tokens. Since $\mathcal{L}_{\text{NTP}}$ cannot guarantee that $h_t$ converges to $h_t^*$ (optimal hidden state), $h_t$ can be misplaced on another geodesic. This leads to mode collapse (the red dotted line mistakenly continues the generation, misattributing Hinton's Nobel Prize to an arbitrary person, or if the error deviates in the opposite direction and precludes a winner).

be modeled as a Stochastic Differential Equation (SDE) with a Brownian motion term.

## 3. Semantic Tube Prediction

A key challenge in minimizing the error $\epsilon_t$ is that the optimal trajectory $h^*$ remains latent and unknown. To address this, we must postulate a structural property that allows us to estimate $h^*$, leading us to the Geodesic Hypothesis. In this section, we formalize this hypothesis and subsequently introduce Semantic Tube Prediction (STP).

### 3.1. Semantic Tube

If the Principle of Least Action holds, the trajectories of the token sequence $x_{\leq t+1}$ in Equation (1) must be geodesics, which are locally linear almost everywhere. Since $h_t^* = \mathring{f}(x_{\leq t})$, when $\mathring{f}(\cdot)$ is smooth enough, $h_t^*$ is also expected to be locally linear almost everywhere. Hence the Geodesic Hypothesis:

*The trajectory of $x_{\leq t} \in \mathbb{R}^{T \times d_{\text{model}}}$ is locally linear almost everywhere. Similarly, the trajectory $h_t - \epsilon_t \in \mathbb{R}^d$ is locally linear almost everywhere.*

We first formally define local linearity. Subsequently, we demonstrate that the Semantic Tube compresses the trajectory $h_t$ within a tube centered at $h_t^*$.

**Definition 1** (Local Linearity). A time-indexed trajectory

$h^*$ is defined as locally linear if $\exists \tau, \exists \varepsilon$ such that for any time indices $s < r < t$ satisfying $|t - s| \leq \tau$, we have:

$$\|(h_r^* - h_s^*)_{\perp h_t^* - h_s^*}\|_2 \leq \varepsilon \tag{4}$$

where $x_{\perp y}$ denotes the component of vector $x$ that is perpendicular to vector $y$.

Definition 1 captures the intuition that if a trajectory is locally linear, each local segment can be approximated by a straight line connecting its endpoints.

Next, we demonstrate that the Semantic Tube forces $h$ to approximate $h^*$.

**Lemma 1** (Straightening Lemma). *If $h_s = h_s^*$, $h_t = h_t^*$, and $\mathcal{L}_{\mathrm{STP}} \leq \epsilon$ for all $r$ satisfying $s < r < t$, then*

$$\|(h_r - h_s)_{\perp h_t^* - h_s^*}\|_2 \leq \sqrt{2\epsilon}\|h_r - h_s\|_2.$$

Proof is deferred to Section D.

Let $\|h_r - h^*\|_2 = \min_{r'} \|h_r - h_{r'}^*\|_2$ denote the minimum distance from $h_r$ to the trajectory $h^*$. We establish the following theorem:

**Theorem 1** (Semantic Tube). *If $h^*$ is locally linear and for all $r$ satisfying $0 \leq s < r < t \leq \tau$, $\mathcal{L}_{\mathrm{STP}} \to 0$, then*

$$\|h_r - h^*\|_2 \lesssim \varepsilon + \delta$$

*where $\delta = \max(\|h_s - h_s^*\|_2, \|h_t - h_t^*\|_2)$*

*Proof Sketch.* First assume $h_s = h_s^*$ and $h_t = h_t^*$. In this scenario, $\|h_r - h_s\|_2 = \|h_r - h_s^*\|$. Applying the triangle inequality yields $\|h_r - h_s^*\| \leq \|h_r^* - h_s^*\|_2 + \epsilon_r$. Notice $h_r^*$ and $h_s^*$ are fixed, by Lemma 1, $\|(h_r - h_s)_{\perp h_s^* - h_s^*}\|_2 \to 0$. By Definition 1 and the triangle inequality, it follows that $\|h_r - h^*\|_2 \lesssim \varepsilon$.

For general case, note that the distance between line segment $[h_s, h_t]$ and the optimal segment $[h_s^*, h_t^*]$ is at most $\delta$, the total error bound is $\varepsilon + \delta$. □

In LLMs, it is standard to assume all sequences begin with `<bos>` and end with `<eos>`; thus, it is reasonable to assume that at the boundary, $\max(\|h_s - h_s^*\|_2, \|h_t - h_t^*\|_2)$ is small enough. Section E provides a construct under which we can safely assume that $\delta$ is sufficiently small.

In practice, the indices $s < r < t$ are selected randomly. Consequently, minimizing $\mathcal{L}_{\mathrm{STP}}$ effectively drives $\mathbb{E}[1 - \cos(h_t - h_r, h_r - h_s)] \to 0$. By Markov's inequality, for any $\epsilon$, $P(1 - \cos(h_t - h_r, h_r - h_s) > \epsilon) \to 0$. This leads to the following corollary:

**Corollary 1** (Random Tube). *For randomly selected $s < r < t$, if $\mathcal{L}_{\mathrm{STP}} \to 0$, then for any $\epsilon$,*

$$P(\|h_r - h^*\|_2 > \varepsilon + \epsilon) \to 0$$

Corollary 1 implies that if $\mathcal{L}_{\mathrm{STP}} \to 0$ for a given sequence, then with high probability, the trajectory of the sequence's hidden states is confined within a tube centered around the optimal trajectory $h^*$.

However, at inference time, the Brownian motion term diverges into a cone whose radius scales as $\propto \sigma_t \sqrt{t}$, see Section F for details.

### 3.2. Rotary Positional Embedding (RoPE)

This section formalizes the ODE assumption with RoPE (Su et al., 2024).

Let $x_{\leq t}$ be the vector representation of the subsequence up to token $t$. The query and key representations are obtained via linear projections $Q = P_Q x_{\leq t}$ and $K = P_K x_{\leq t}$. Let $R_i$ be the rotation matrix for position $i$, and let $q_i$ and $k_i$ the corresponding query and key column vectors for the $i$-th token. Under RoPE, the attention score between token $i$ and $j$ is computed as $\mathrm{score}(i, j) = (R_i q_i)^\top (R_j k_j) = q_i^\top R_i^\top R_j k_j$ for $i, j \leq t$. Hence the attention score matrix is fully determined by the sequence representations $x_{\leq t}$ and the current positional boundary $t$. The remainder of the transformer network involves matrix multiplication with the values $V = P_V x_{\leq t}$ and applying the unembedding function $u(\cdot)$, which remain functions of $x_{\leq t}$ and $t$. Hence, our ODE assumption is compatible with RoPE.

### 3.3. Practical Considerations

Since the forward pass naturally computes $h_s$, $h_r$, and $h_t$, the STP loss introduces negligible computational overhead—primarily the cost of computing cosine similarity. This is significantly more efficient than the fractional extra forward passes required by LLM-JEPA (Huang et al., 2025). Furthermore, because indices $s$, $r$, and $t$ can be selected randomly, STP eliminates the need for manual scaffolding of a two-view structure. In summary, STP effectively addresses the two primary limitations that have hindered the broader adoption of LLM-JEPA. Additionally, STP avoids the complexity of a predictor network (often a requirement in LLM-JEPA), as local linearity implies an identity predictor. Like LLM-JEPA, the STP loss is applied exclusively during training and is not required at inference time.

Further implementation details are provided in Section G.

## 3.4. Related Work

Our approach addresses the classic **Exposure Bias** problem (Bengio et al., 2015), originally identified in recurrent neural networks (RNNs) (Elman, 1990; Siegelmann & Sontag, 1995). The problem arises because the model is trained with **Teacher Forcing** (Williams & Zipser, 1989)—conditioning on the ground-truth history—but must rely on its own potentially drifting predictions during inference. Although Maximum Likelihood Estimation ($\mathcal{L}_{\mathrm{NTP}}$ in the case of LLMs) is empirically effective, Huszár (2015) argues that it optimizes an objective different from generation quality, motivating our combined loss $\mathcal{L}_{\mathrm{NTP}} + \mathcal{L}_{\mathrm{STP}}$.

**JEPAs** (Assran et al., 2023; Baevski et al., 2022) learn predictive representations across views, offering theoretical benefits (Littwin et al., 2024) despite the risk of dimensional collapse (Jing et al., 2021; Kenneweg et al., 2025). While recent works extend these objectives to LLMs (Barrault et al., 2024; Wang & Sun, 2025), LLM-JEPA (Huang et al., 2025) is bottlenecked by manual two-view scaffolding and the computational cost of additional forward passes, neither is a problem for $\mathcal{L}_{\mathrm{STP}}$.

Our framework extends the philosophy of **Energy-Based Models** (EBMs) (LeCun et al., 2006), which learn to assign low energy to compatible configuration of variables. While EBMs and recent architectures like JEPA (LeCun, 2022) typically minimize energy at specific states, our approach invokes the Principle of Least Action to minimize the action—the integral of the Lagrangian along the generation trajectory. By enforcing geodesic constraints via $\mathcal{L}_{\mathrm{STP}}$, we generalize state-wise (or local) energy minimization to trajectory-wise action minimization, ensuring the generation follows the path of least resistance.

**Scaling Laws** govern the power-law relationship between compute, data, and parameters in both pre-training (Kaplan et al., 2020; Hoffmann et al., 2022) and fine-tuning (Zhang et al., 2024). While recent data efficiency research emphasizes identifying high-density subsets (Sorscher et al., 2022) or synthetic curation (Gunasekar et al., 2023; Muennighoff et al., 2023), $\mathcal{L}_{\mathrm{STP}}$ enhances the training SNR directly, obviating the need for explicit data subset selection.

**SDE/ODE Perspective**: Kong et al. (2020) interpreted ResNets as "Neural SDEs" which has a Brownian motion term. While Tong et al. (2025) recently adapted ODEs for LLMs, they model evolution across network depth (layers). Our work takes an orthogonal approach, focusing instead on the temporal dynamics of hidden states across the token sequence.

**The Linear Representation Hypothesis** (LRH) (Park et al., 2024; 2025) posits that simple concepts are encoded as directions in representation space. The Geodesic Hypothesis is conceptually related but distinct: it suggests that both simple and composed concepts, when expressed as token sequences, can evolve along locally linear trajectories in representation space. Under this perspective, the vector arithmetic observed in LRH, such as $\vec{v}_{Paris} - \vec{v}_{France} + \vec{v}_{Italy} \approx \vec{v}_{Rome}$, is consistent with the idea that relational token sequences may trace approximately linear paths, e.g., $(\vec{v}_{Paris}, \vec{v}_{to}, \vec{v}_{France}, \vec{v}_{is}, \vec{v}_{Rome}, \vec{v}_{to}, \vec{v}_{Italy})$ lying close to a low-curvature trajectory; see Figure 3. Thus, we view the two hypotheses as offering complementary geometric perspectives on how conceptual relations may be represented.

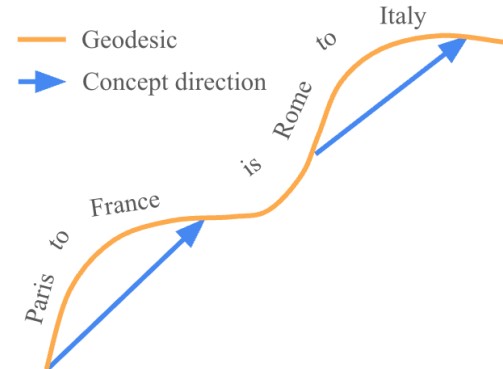

*Figure 3*. When the sentence aligns on a geodesic, the concept direction naturally aligns.

**The Manifold Hypothesis** (Kiani et al., 2024; Robinson et al., 2025; Whiteley et al., 2025) posits that learned representations tend to concentrate on a comparatively simple and smooth manifold. The Geodesic Hypothesis is conceptually aligned with this view, but emphasizes the geometry of transitions between representations: it suggests that token-level representations may evolve along low-curvature trajectories in representation space. From this perspective, manifold-like structure and locally geodesic dynamics provide complementary descriptions of representation geometry, rather than one serving as a formal consequence of the other.

**The Curvature Straightening Phenomenon** (Hosseini & Fedorenko, 2023; Hénaff et al., 2021) observes that the training process tends to straighten the curvature between consecutive tokens. We interpret this as a manifestation of the underlying geodesic, which approximates a straight line.

The **Neural Tangent Kernel (NTK)** simplifies infinite-width dynamics (Jacot et al., 2018), a framework generalized to Transformers (Hron et al., 2020; Yang & Littwin, 2021) and compatible feature learning regimes (Yang & Hu, 2021). While Seleznova & Kutyniok (2022) note the importance of the depth-to-width ratio, modern LLMs typically operate in the requisite width $\gg$ depth regime.

The application of **geodesic geometry to LLMs** remains underexplored, with existing studies primarily restricted to interpolating representations across models (Deng et al.,

2025; Yu et al., 2024).

# 4. Experiments

We conduct extensive experiments to show the performance of Semantic Tube across models, datasets, and model sizes. We also show that accuracy barely budges when the training dataset is halved. Both accuracy and data efficiency are solid evidence that Semantic Tube improves SNR. We ablate on various setups, including LLM-JEPA style explicit two-views and curvature straightening. Lastly we show how to tune $\lambda$ in practices.

Implementing $\mathcal{L}_{\mathrm{STP}}$ is straightforward with HuggingFace `transformers`. When computing loss, we grab per-token `hidden_state` $h$ from last layer, pick (random) indices $s < r < t$, and compute $1 - \cos(h_t - h_r, h_r - h_s)$. Across all experiments, we follow LLM-JEPA (Huang et al., 2025) to pick 5 random seeds: 82, 23, 37, 84, and 4, and report both mean accuracy and standard deviation. This also allows us to report $p$-value of paired, single-tailed $t$-Test. We inherit optimal number of epochs and learning rate from LLM-JEPA. $\lambda$ is separately tuned.

To ensure a fair comparison between NTP and STP, we adopt the same hyperparameter tuning protocol from LLM-JEPA (Huang et al., 2025). We first establishes the optimal learning rate for vanilla fine-tuning, and subsequently fixes that rate while tuning auxiliary parameters (like $\lambda$). While this means we likely did not utilize the true optimal learning rate for STP, it establishes a rigorous lower bound: it proves that STP outperforms vanilla fine-tuning even when subjected to conditions optimized specifically for the vanilla baseline.

## 4.1. Loss Landscape

We begin by analyzing the loss landscape by fine-tuning Llama-3.2-1B-Instruct (Grattafiori et al., 2024) on the NL-RX-SYNTH (Locascio et al., 2016) dataset.

Figure 4(a) demonstrates that in regular fine-tuning, minimizing $\mathcal{L}_{\mathrm{NTP}}$ does not automatically minimize $\mathcal{L}_{\mathrm{STP}}$. With the Semantic Tube, however, $\mathcal{L}_{\mathrm{STP}}$ continues to decrease even after $\mathcal{L}_{\mathrm{NTP}}$ plateaus, corroborating (P1). Moreover, while $\mathcal{L}_{\mathrm{NTP}}$ remains comparable between regular and Semantic Tube fine-tuning, there is a significant gap in $\mathcal{L}_{\mathrm{STP}}$. This confirms that the SNR gain is driven by $\mathcal{L}_{\mathrm{STP}}$, validating the analysis in Section 2.2 that $\mathcal{L}_{\mathrm{NTP}}$ alone is insufficient for generation quality and that $\mathcal{L}_{\mathrm{STP}}$ acts as a necessary complement.

Figure 4(b) illustrates that increasing $\lambda$ on a logarithmic scale reduces $\mathcal{L}_{\mathrm{STP}}$ linearly across a wide range, while $\mathcal{L}_{\mathrm{NTP}}$ remains stable. Given $\mathcal{L}_{\mathrm{STP}} = 1 - \cos(h_t - h_r, h_r - h_s)$, a value of $\mathcal{L}_{\mathrm{STP}} > 1.0$ implies that the trajectory vector

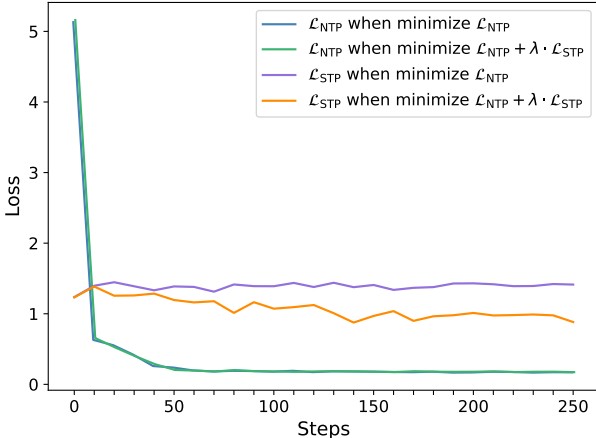

*(a)* Loss curve

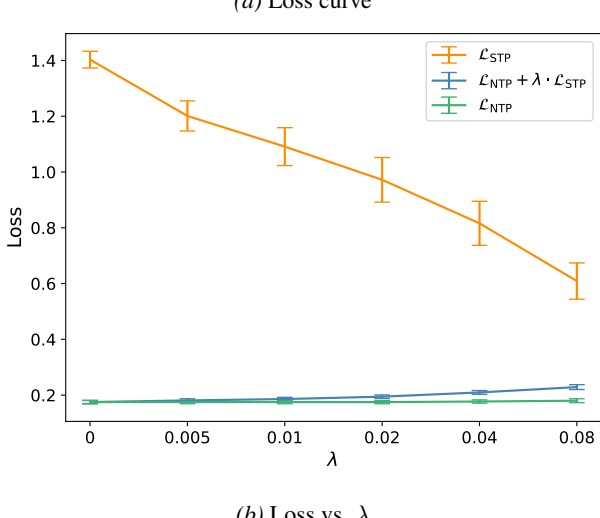

*(b)* Loss vs. $\lambda$

*Figure 4.* Loss landscape. **(a)** When $\mathcal{L}_{\mathrm{NTP}}$ plateaus, $\mathcal{L}_{\mathrm{STP}}$ continues to decrease. Furthermore, minimizing $\mathcal{L}_{\mathrm{NTP}}$ does not automatically minimize $\mathcal{L}_{\mathrm{STP}}$. **(b)** Across a wide range of $\lambda$, increasing $\lambda$ on a logarithmic scale reduces $\mathcal{L}_{\mathrm{STP}}$ linearly, while $\mathcal{L}_{\mathrm{NTP}}$ remains unchanged.

$h_t - h_r$ diverges significantly (essentially reversing direction) relative to $h_r - h_s$. At $\lambda = 0$ (regular fine-tuning), $\mathcal{L}_{\mathrm{STP}} \approx 1.4$ indicates a trajectory resembling erratic Brownian motion. At $\lambda = 0.08$, $\mathcal{L}_{\mathrm{STP}}$ drops to 0.6, reflecting a substantially smoother path. Notably, while the optimal performance is achieved at $\lambda = 0.02$ (Table 3), the accuracy at $\lambda = 0.08$ is only marginally lower (Figure 7).

## 4.2. Better Accuracy

**On Various Datasets**: We first fine-tune Llama-3.2-1B-Instruct to demonstrate that Semantic Tube yields significant accuracy improvements over regular fine-tuning and LLM-JEPA across diverse datasets: NL-RX-SYNTH, NL-RX-TURK (Locascio et al., 2016), GSM8K (Cobbe et al., 2021), Spider (Yu et al., 2018), NQ-Open (Lee et al., 2019), and

HellaSwag (Zellers et al., 2019). Figure 5(a) illustrates the superior performance of Semantic Tube compared to regular fine-tuning and LLM-JEPA.

**On Various Model Families**: Next, we extend our evaluation to various model families. In addition to Llama, we evaluate gemma-2-2b-it (Team et al., 2024), OpenELM-1.1B-Instruct (Mehta et al., 2024), and OLMo-2-0425-1B-Instruct (OLMo et al., 2024) on NL-RX-SYNTH, as well as Qwen3-1.7B (Yang et al., 2025) and DeepSeek-R1-Distill-Qwen-1.5B (DeepSeek-AI et al., 2025) on GSM8K. The results are presented in Figure 5(b).

**On Various Model Sizes**: Finally, we examine scalability across model sizes using Llama-3 1B, 3B, and 8B models. Results are shown in Figure 5(c).

### 4.3. Data Efficiency

Data efficiency is another crucial metric demonstrating improved SNR. We randomly select subsets of $\frac{1}{2}$, $\frac{1}{4}$, $\frac{1}{8}$, $\frac{1}{16}$, and $\frac{1}{32}$ of the NL-RX-SYNTH dataset and perform both Semantic Tube and regular fine-tuning on Llama-3 1B, 3B, and 8B models. To compensate for the reduced number of training steps, we scale the epochs proportionally: with a $\frac{1}{n}$ dataset fraction, we run $n\times$ epochs. For Semantic Tube, accuracy shows a negligible drop when the training dataset is halved and remains robust until the dataset is reduced to $\frac{1}{16}$, at which point it matches the accuracy of regular fine-tuning on the full dataset. In contrast, regular fine-tuning suffers a significant drop immediately when the dataset is halved. See Figure 1 for 1B results and Figure 12 for 3B and 8B results.

We also experimented with half compute ($\frac{n}{2}\times$ epochs) combined with a $2\times$ learning rate. In both full and half compute scenarios, we also tested $2 \times \lambda$. Interestingly, although the half-compute, double-learning-rate setting does not yield optimal accuracy at $\frac{1}{2}$ or full training data, it performs comparatively better when the dataset fraction is $< \frac{1}{2}$.

The improved accuracy and data efficiency provide strong evidence that Semantic Tube improves SNR (see Section H for formal proofs linking SNR to accuracy and data efficiency). This validates (P2) and supports the proposed noise/signal decomposition in Figure 1, where the component perpendicular to the tube represents noise. Consequently, it supports the hypothesis that the geodesic is locally linear; otherwise, it could not be effectively approximated by the tube.

In addition to NL-RX-SYNTH, we conduct the same data-efficiency experiments on HellaSwag and observe a similar trend, as shown in Table 1.

We further evaluate whether STP can benefit pretraining from random initialization. We initialize a model from

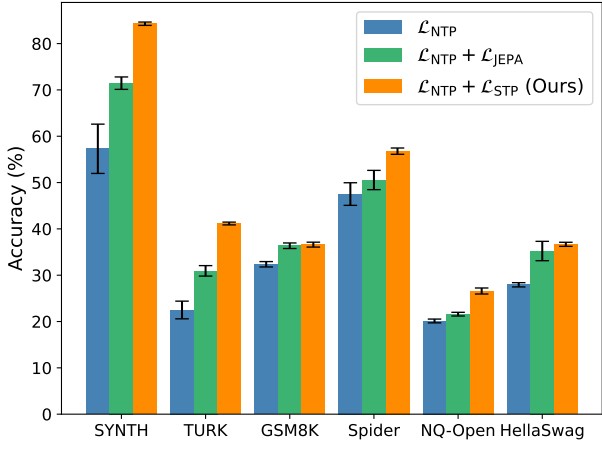

*(a)* Datasets

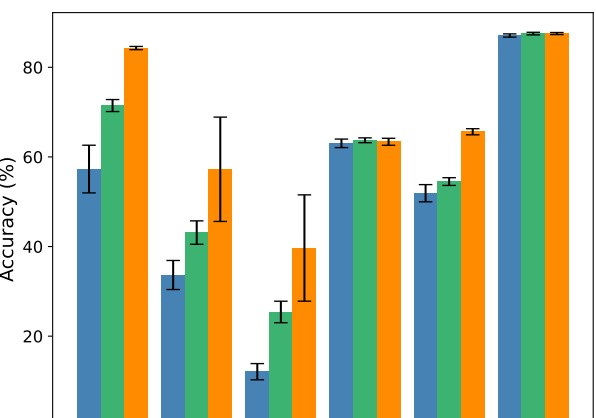

*(b)* Model families

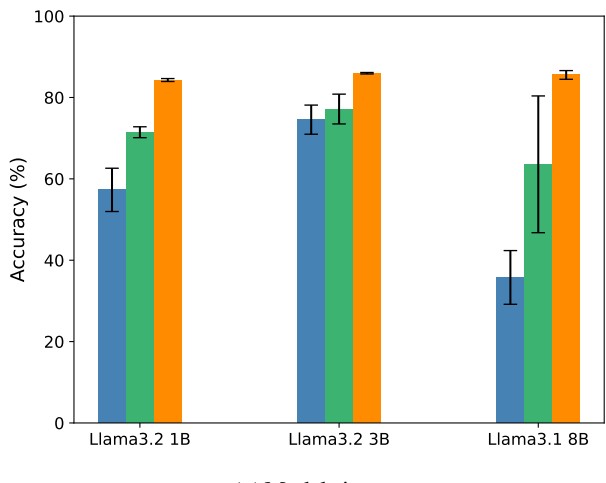

*(c)* Model sizes

*Figure 5.* Semantic Tube ($\mathcal{L}_{\mathrm{NTP}} + \mathcal{L}_{\mathrm{STP}}$, our approach) demonstrates superior performance across **(a)** datasets, **(b)** model families, and **(c)** model sizes compared to regular fine-tuning ($\mathcal{L}_{\mathrm{NTP}}$) and LLM-JEPA ($\mathcal{L}_{\mathrm{NTP}} + \mathcal{L}_{\mathrm{JEPA}}$).

*Table 1.* Data-efficiency results on HellaSwag. STP remains substantially more data-efficient than the regular training baseline.

| STP | 1/2 | 1/4 | 1/8 | 1/16 | 1/32 | NTP |
|---|---|---|---|---|---|---|
| 36.67 | 34.26 ±0.41 | 32.14 ±0.39 | 31.53 ±0.87 | 30.39 ±0.72 | 29.23 ±0.54 | 27.93 |

scratch and pretrain it on HellaSwag by concatenating each context with its correct completion. Evaluation is performed on a held-out set in cloze format: for each example, we compute the negative log-likelihood (NLL) of each candidate completion and select the option with the lowest NLL. Despite the highly constrained setting of only 4M pretraining tokens for a single epoch, adding STP yields a statistically significant improvement over regular pretraining, increasing accuracy from $24.90 \pm 0.49$ to $25.26 \pm 0.33$ ($p = 0.0204$). Section M provides our pretraining methodology.

### 4.4. Preserving Diversity

In this section, we demonstrate that Semantic Tube preserves diversity. In the NL-RX-SYNTH dataset, some regular expressions end with ".*", while others end with ".*.*". Although functionally equivalent, these variations represent a nuanced preference by the dataset creator; a robust neural network should be able to learn and preserve this diversity. As shown in Table 2, we find that regular fine-tuning struggles to learn either pattern effectively. LLM-JEPA learns the former pattern well but fails on the latter, likely because the former dominates the training set by a factor of $35\times$. In contrast, Semantic Tube successfully learns both patterns. We list representative samples from the SYNTH dataset ending with either ".*" or ".*.*" in Table 4.

*Table 2.* Accuracy on functionally equivalent regular expression suffixes ".*" and ".*.*". Semantic Tube effectively captures nuanced preferences, whereas LLM-JEPA exhibits mode collapse by biasing towards ".*", which is $35\times$ more prevalent in the training set than ".*.*".

| Suffix | Semantic Tube | Regular | LLM-JEPA |
|---|---|---|---|
| .* | 88.5% | 29.9% | 68.9% |
| .*.* | 68.0% | 28.0% | 32.0% |

Following LLM-JEPA, we compute the singular value decomposition (SVD) of $\text{Enc}(\text{Text}) - \text{Enc}(\text{Code})$ to gain insight into the learned representations. Interestingly, we find (Figure 6) that Semantic Tube exhibits polymorphism: when the difference vectors $\text{Enc}(\text{Text}) - \text{Enc}(\text{Code})$ are normalized, the singular value spectrum aligns with LLM-JEPA; however, without normalization, it closely resembles regular fine-tuning. This indicates that Semantic Tube enforces structure on the directions (normalized vectors) while

tolerating complexity on the raw vectors. We conjecture that this mechanism allows Semantic Tube to maintain flexibility and preserve diversity.

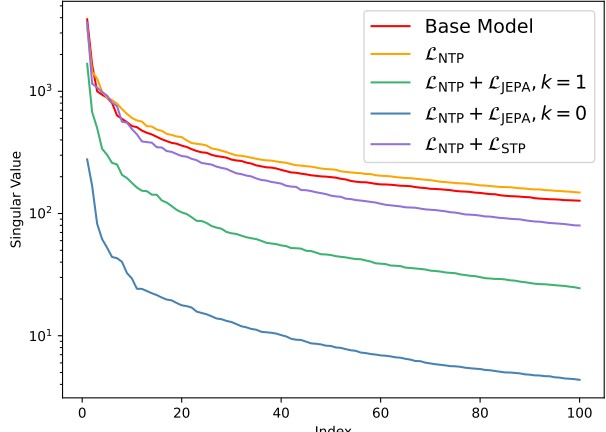

*(a)* Without Normalization

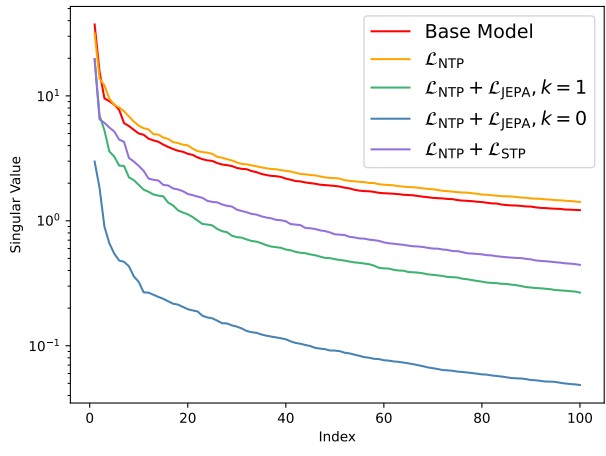

*(b)* With Normalization

*Figure 6.* SVD decomposition demonstrating Semantic Tube's polymorphism. **(a)** Without normalization, the SVD profile closely resembles regular fine-tuning. **(b)** With normalization, the SVD aligns with LLM-JEPA. Collectively, this indicates that Semantic Tube enforces a simple structure on the directions (normalized vectors) mapping Text to Code, while tolerating complexity in the unnormalized vectors. Note that the relative relationships among the base model, regular fine-tuning, and LLM-JEPA remain unchanged with or without normalization.

Collectively, these results validate (P3).

### 4.5. Tuning $\lambda$

Semantic Tube introduces a single hyperparameter, $\lambda$. Empirically, we observe that the accuracy vs. $\lambda$ curve is concave (Figure 7), typically peaking between 0.01 and 0.08 (Table 3). Notably, this behavior persists across other variations (see Section 4.6): the accuracy curves remain concave,

and the optimal $\lambda$ consistently falls within the 0.01–0.08 range (see Figure 13). This validates (P4).

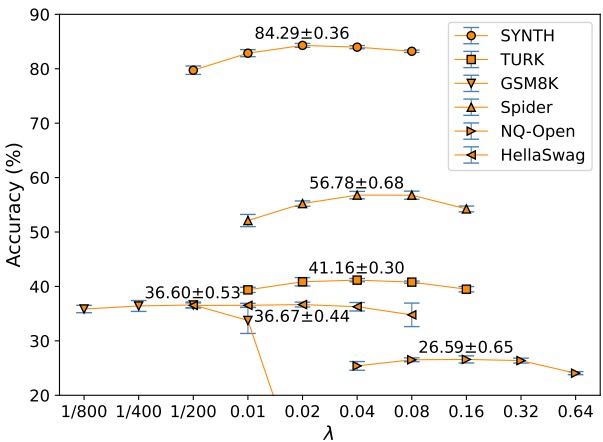

*Figure 7.* Impact of $\lambda$ tuning on Llama-3 1B across various datasets. In most cases, peak performance is achieved within the range of 0.01 to 0.08.

*Table 3.* Optimal $\lambda$ values yielding maximum accuracy.

| SYNTH | TURK | GSM8K | Spider | NQ | HS |
|---|---|---|---|---|---|
| 0.02 | 0.04 | 0.005 | 0.04 | 0.16 | 0.02 |
| Gemma2 | Qwen3 | R1 Dist | OLMo | OpenELM | |
| 0.005 | 0.02 | 0.04 | 0.01 | 0.04 | |
| Llama3 3B | | | Llama3 8B | | |
| 0.01 | | | 0.0025 | | |

### 4.6. Ablation

We conducted extensive ablation studies on design decisions, establishing that $\mathcal{L}_{\mathrm{STP}}$ yields superior performance compared to all variations (Figure 8). Full details are provided in Section L. We specifically note that the **Pred** variant—which trains a linear projector $P$ to minimize $\mathcal{L}_{\mathrm{STP}} = 1 - \cos(P(h_r - h_s), h_t - h_r)$—results in degraded performance in all configurations. This validates (P5).

## 5. Conclusion and Future Work

This paper proposes the Geodesic Hypothesis, which posits that token sequence trajectories on the LLM manifold are locally linear geodesics. Based on it, we introduce Semantic Tube Prediction (STP)—a learning objective complementary to Next Token Prediction—which compresses hidden state trajectories into a signal-rich tube centered on the geodesic. Our approach generalizes LLM-JEPA by eliminating the need for manual scaffolding of two-view structures, additional compute, or auxiliary predictors. Empirically,

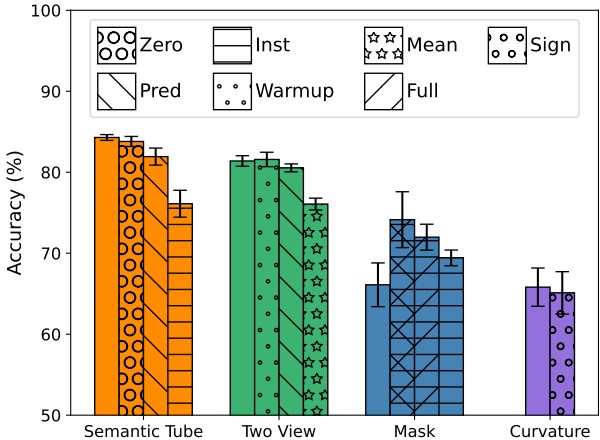

*Figure 8.* Ablation study. Semantic Tube (our approach) outperforms all variations. Within the Semantic Tube family, alternative configurations consistently degrade performance.

STP significantly improves Signal-to-Noise Ratio, allowing models to maintain accuracy even when training data is reduced to $\frac{1}{16}$, thereby challenging standard Power Law scaling. Our framework unifies the Linear Representation and Manifold Hypotheses under the Principle of Least Action.

**Future Work.** Several directions remain open. First, STP may provide a useful training-time complement to test-time scaling methods such as Chain-of-Thought prompting and budget-forcing approaches like s1 (Muennighoff et al., 2025). By regularizing hidden-state trajectories to remain locally coherent within the semantic tube, STP may improve the structural consistency of generated reasoning traces, while test-time methods control how much reasoning is produced. Second, applying STP during large-scale pretraining is an important next step. Our results suggest that STP can improve data efficiency in fine-tuning and small-scale training settings, but its impact on longer-horizon pretraining dynamics, scaling behavior, and representation formation remains to be systematically studied. Third, extending STP beyond text-only LLMs to multimodal models and vision-language settings (Radford et al., 2021; Alayrac et al., 2022) is a natural direction, especially in light of recent JEPA-style vision-language architectures. Finally, STP may improve model steerability (Subramani et al., 2022; Turner et al., 2023): if concepts are represented by approximately linear directions, then locally linear hidden trajectories should yield cleaner and more robust activation-steering vectors. These applications require models to be pretrained or fine-tuned with the STP objective, since standard pretrained models do not necessarily exhibit low STP loss. We leave systematic evaluations of STP for test-time scaling, pretraining, multimodal learning, and activation engineering to future work.

## Impact Statement

This paper presents work whose goal is to advance the field of machine learning. There are many potential societal consequences of our work, none of which we feel must be specifically highlighted here.

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

## A. Training ODE

In this section, we present a form of $u(\cdot)$ and $f(\cdot)$ such that $x_{\leq t+1} \ominus x_{\leq t} = x_{\leq t+1} - x_{\leq t}$. Throughout the section, we slightly abuse notation by letting $x_t$ denote both a token and its embedding vector $x_t \in \mathbb{R}^{d_{\text{model}}}$, and letting $x_{\leq t}$ denote both a token sequence and its embedding vector $x_{\leq t} \in \mathbb{R}^{T \times d_{\text{model}}}$:

$$x_{\leq t} = [x_1, ..., x_t, 0, ..., 0].$$

Let $f(x_{\leq t}) \in \mathbb{R}^d$. Let $u(\cdot) : \mathbb{R}^d \to \mathbb{R}^{d_{\text{model}}}$ be the unembedding function that maps the hidden state back to the token embedding.

Note that we need a function to lift $u(f(x_{\leq t}))$ from $\mathbb{R}^{d_{\text{model}}}$ to $\mathbb{R}^{T \times d_{\text{model}}}$. Define $v(\cdot, \cdot) : \mathbb{R}^{d_{\text{model}}} \times \mathbb{N} \to \mathbb{R}^{T \times d_{\text{model}}}$ such that

$$v(x, t) = [0, ..., 0, \underbrace{x}_{\text{index } t+1}, 0, ..., 0]$$

Hence, we have

$$x_{\leq t+1} = v(u(f(x_{\leq t})), t)$$

Define the $\ominus$ operator as

$$x_{\leq t+1} \ominus x_{\leq t} = v(x_{t+1}, t)$$

By the definition of $v(\cdot, \cdot)$, we have

$$x_{\leq t+1} \ominus x_{\leq t} = x_{\leq t+1} - x_{\leq t}$$

Note that the network is now in the form $v(u(f(x_{\leq t})), t)$, which can be written as $g(x_{\leq t}, t)$ and satisfies the formulation of an ODE.

## B. Inference SDE

At training time, the unembedding error $\epsilon_t$ does not propagate to the next token. However, at inference time, $h_{t+1}$ depends (indirectly) on $h_t$, causing $\epsilon_t$ to accumulate into a Brownian motion term.

Yang & Littwin (2021) established that in the limit of infinite width, the pre-activations of a neural network (and thus the hidden state) are well-approximated by Gaussian processes. Hence, we can assume $\epsilon_t$ are i.i.d. Gaussian. Furthermore, as shown by (Yang & Littwin, 2021), $\epsilon_t$ remains i.i.d. Gaussian when passed through a randomly initialized neural network, which remains constant in the infinite-width limit. Consequently, $\epsilon_t$ accumulates to form a Brownian motion term $dW_t$. Thus the inference process can be modeled by a Stochastic Differential Equation (SDE).

**Proposition 2** (Inference SDE). *The inference process of an LLM can be modeled by an SDE in the token sequence space* $\mathbb{R}^{T \times d_{\text{model}}}$,

$$dx_{\leq t} = \mathring{u} \circ \mathring{f}(x_{\leq t}) dt + \sigma_t dW_t$$

Consider the example in Figure 2, if the Brownian motion shifts the top trajectory to the bottom, mode collapse occurs. Conversely, if the bottom trajectory shifts to the top, mode collapse occurs. This motivates the construction of an approach to explicitly suppress $\epsilon_t$. Indeed, Section 4.1 demonstrates that next token prediction alone is insufficient for high-quality generation, making our approach a necessary complement.

## C. Context-Aware Hidden State

We can view $h_t - h_s$ as the semantic evolution induced by the sub-sequence $x_{[s,t]}$ given the context $x_{\leq s}$. In this sense, $h_t - h_s$ acts as a context-aware hidden state transition, which is significantly more informative than the static hidden state of the isolated sub-sequence $x_{[s,t]}$.

For example, given the prefix $\vec{v}_{\text{The}}, \vec{v}_{\text{capital}}, \vec{v}_{\text{of}}$, appending the token $\vec{v}_{\text{France}}$ shifts the overall semantic trajectory toward $\vec{v}_{Paris}$. However, given a different prefix $\vec{v}_{\text{The}}, \vec{v}_{\text{language}}, \vec{v}_{\text{of}}$, appending the same token $\vec{v}_{France}$ shifts the trajectory toward $\vec{v}_{French}$. If we were to compute the hidden state of $\vec{v}_{France}$ in isolation, we would lose this contextual nuance and fail to capture the context-specific semantic semantic shift.

Thus, $h_t - h_s$ serves as a context-aware representation of the added information.

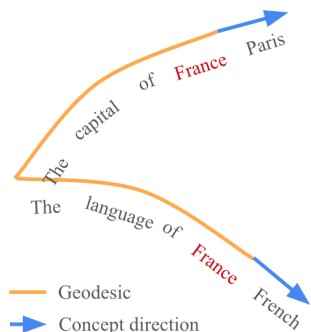

*Figure 9.* The same token $\vec{v}_{France}$ directs the geodesic along different concept directions when appended to distinct prefixes, illustrating the necessity of the context-aware state difference $h_t - h_s$.

## D. Proof of the Straightening Lemma

In this section, we provide the proof for Lemma 1. The objective is to show

$$\|(h_r - h_s)_{\perp h_t^* - h_s^*}\|_2 \le \sqrt{2\epsilon}\|h_r - h_s\|_2.$$

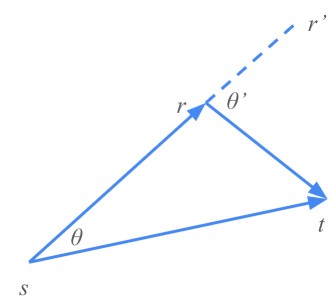

*Figure 10.* Geometric illustration for the proof of Lemma 1

Referring to Figure 10, we have

$$\|(h_r - h_s)_{\perp h_t^* - h_s^*}\|_2 = \|h_r - h_s\|_2 \cdot \sin\theta$$

Since $\theta' \ge \theta$, it follows that

$$\|(h_r - h_s)_{\perp h_t^* - h_s^*}\|_2 \le \|h_r - h_s\|_2 \cdot \sin\theta'$$

We also have

$$\mathcal{L}_{\text{STP}} = 1 - \cos\theta' \le \epsilon$$

When $\epsilon$ is sufficiently small, we can approximate $\cos\theta' \approx 1 - \frac{\theta'^2}{2}$. Hence

$$\frac{\theta'^2}{2} \lesssim \epsilon$$

Rearranging gives

$$\theta' \lesssim \sqrt{2\epsilon}$$

Also, when $\theta'$ is sufficiently small, $\sin\theta' \approx \theta'$. Therefore

$$\|(h_r - h_s)_{\perp h_t^* - h_s^*}\|_2 \le \|h_r - h_s\|_2 \cdot \sin\theta' \lesssim \sqrt{2\epsilon}\|h_r - h_s\|_2. \qquad \square$$

## E. Proof of the Semantic Tube Theorem

We introduce two auxiliary tokens, `<before-bos>` and `<after-eos>`. The token `<before-bos>` appears only at the 0-th position and always precedes `<bos>`, while `<after-eos>` appears only at the $\tau + 1$-th position and always follows `<eos>`. This augmentation increases the total sequence length from $\tau$ to $\tau + 2$. By anchoring the sequence with `<before-bos>` and `<after-eos>`, we ensure that the boundary conditions $h_0 = h_0^*$ and $h_{\tau+1} = h_{\tau+1}^*$ are satisfied.

The proof follows from these conditions. □

## F. Inference Cone

As STP explicitly reduces $\epsilon_t$, it lowers $\sigma_t$ in the Brownian Motion term of Proposition 2. At inference time, the Brownian motion term causes the token sequence trajectory diverge into a cone whose radius grows at a rate $\propto \sigma_t \sqrt{t}$. A lower $\sigma_t$ reduces the probability that the cone collides with another token sequence, which would causes mode collapse (Figure 11).

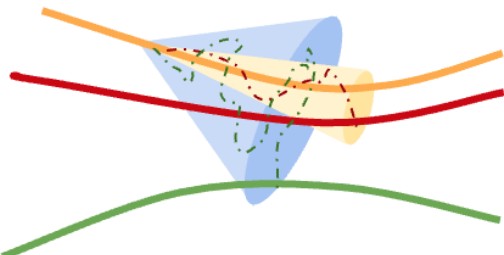

*Figure 11.* The inference cone defines the probabilistic range of a Brownian motion, and its radius grows $\propto \sigma_t \sqrt{t}$. A larger $\sigma_t$ leads to a wider cone, which has a high probability of colliding with a token sequence trace that is far away (blue cone and green geodesic), while a smaller $\sigma_t$ leads to a narrower cone that may only collide with a nearby trace (yellow cone and red geodesic). The dotted red and green fine lines are the Brownian motions confined by the yellow and blue cones, respectively.

**Proposition 3** (Inference Cone)**.** *The distortion between $h_t$ and $h_t^*$ behaves as a Gaussian process, where the scale of the deviation grows as $\|h_t - h_t^*\|_2 \propto \sigma\sqrt{t}$*

*Proof.* According to Proposition 2, at inference time, we model the token sequence trajectory as following an SDE $dx_{\leq t} = \mathring{u} \circ \mathring{f}(x_{\leq t})dt + \sigma_t dW_t$, where $\sigma_t dW_t$ is a Brownian motion. Let $h_t = \mathring{f}(x_{\leq t})$ be the hidden state. Let $x_{\leq t}^*$ be the error-free generation satisfying $dx_{\leq t}^* = \mathring{u} \circ \mathring{f}(x_{\leq t}^*)dt$, and let $h_t^* = \mathring{f}(x_{\leq t}^*)$ be the error-free hidden state. We can quantify the distortion between $h_t$ and $h_t^*$ by examining how the Brownian motion is transformed by $\mathring{f}$.

Yang & Littwin (2021) establishes that in the infinite-width limit, $\mathring{f}$ converges to a Neural Tangent Kernel (NTK) determined by random initialization. It further showed that Gaussian noise remains Gaussian when passed through a randomly initialized network. Hence, a Brownian motion remains a Brownian motion when passed through $\mathring{f}$. Therefore,

$$h_t - h_t^* = \sum_{s \leq t} \epsilon_s$$

where $\epsilon_s$ are Gaussian noises. By Donsker's theorem, when $t \to \infty$, $\frac{1}{\sqrt{t}} \sum_{s \leq t} \epsilon_s \sim N(0, \Sigma)$. Consequently, the magnitude of the distortion scales as

$$\left\| \sum_{s \leq t} \epsilon_s \right\|_2 \propto \sigma\sqrt{t}.$$

Putting everything together, the distortion between $h_t$ and $h_t^*$ satisfies $\|h_t - h_t^*\|_2 \propto \sigma\sqrt{t}$ □

Proposition 3 implies that with high probability, the trajectory of the generated hidden state $h$ is confined within a cone centered at $h^*$ whose radius grows at a rate $\propto \sigma\sqrt{t}$.

When mode collapse occurs at inference time, i.e., a generated sequence $x_{\leq t}$ collides with $y_{\leq t'}$, then their corresponding hidden states $h$ and $g$ must collide. Let $\|h^* - g^*\|_2$ be the minimum distance between $h^*$ and $g^*$. By Proposition 3, $\forall \varepsilon > 0$,

$\exists c,$

$$P(\|h^* - g^*\|_2 > c \cdot \sigma \sqrt{t}) \leq \varepsilon$$

On the other hand, $\mathcal{L}_{\text{STP}}$ suppresses $\epsilon_t$ and consequently reduces $\sigma$, which decreases the lower bound of the probability of mode collapse.

## G. Implementation Details

If the training data already possesses a two-view structure, such as a $(query, answer)$ pair, one can leverage it by anchoring $s$ at the beginning of the $query$ and $t$ at the end of the $answer$. However, we suggest that $r$ should be randomly selected to maximize the benefit of the STP loss. As demonstrated in our ablation study, fixing $r$ at the end of the $query$ yields lower accuracy.

Typically, $h_t - h_s$ does not equal the hidden state of the isolated sub-sequence $x_{[s,t]}$. However, as discussed Section C, we can view $h_t - h_s$ as the semantic evolution induced by the sub-sequence $x_{[s,t]}$ given the context $x_{\leq s}$. In this sense, $h_t - h_s$ acts as a context-aware hidden state, which is significantly more informative than the hidden state of $x_{[s,t]}$ computed in isolation. For example, given the prefix $\vec{v}_{\text{The}}, \vec{v}_{\text{capital}}, \vec{v}_{\text{of}}$, appending the token $\vec{v}_{\text{France}}$ shifts the overall meaning to $\vec{v}_{\text{Paris}}$. Conversely, given the prefix $\vec{v}_{\text{The}}, \vec{v}_{\text{language}}, \vec{v}_{\text{of}}$, appending $\vec{v}_{\text{France}}$ shifts the meaning to $\vec{v}_{\text{French}}$. Computing the hidden state of $\vec{v}_{\text{France}}$ separately loses this context and fails to capture the context-specific meaning of the tokens (see Figure 9).

We can also leverage $h_t - h_s$ to bypass unwanted tokens. For example, setting $s > 0$ allows us to skip the system prompt. Similarly, in multiple-choice Q&A, distractor choices that are semantically inconsistent with the $query$ are often located between the $query$ and the correct $answer$. In such cases, we can pick $r$ and $r'$ such that $x_{[s,r]}$ is the $query$ and $x_{[r',t]}$ is the correct $answer$, computing the STP loss as:

$$\mathcal{L}_{\text{STP}} = 1 - \cos(h_t - h'_r, h_r - h_s).$$

This formulation effectively skips the irrelevant choice branches in the middle.

Finally, the STP loss assumes that $h_s$, $h_r$, and $h_t$ are collinear, which may not hold strictly in reality as geodesics can exhibit curvature. In practice, this implies that we must select a small $\lambda$ to tolerate the angular deviation between $h_t - h_r$ and $h_r - h_s$. Indeed, our experiments consistently show that $\lambda \approx 0.01$ is effective across various models, datasets, and model sizes.

## H. Signal-to-Noise Ratio

Directly measuring Signal-to-Noise Ratio (SNR) in the latent representations of LLMs is intractable. In self-supervised learning, the decomposition of activations into "semantic signal" and "nuisance noise" is not explicitly observable without access to the ground-truth data manifold.

In this subsection, we formally show an information theoretic link between SNR and data efficiency and training accuracy. Hence we can validate our hypothesis via the predicted impact on them.

We model LLM training process as extracting information about a discrete target $Y$ (tokens) from continuous latent representations $X$ (hidden states). Let $Y \in \mathcal{V}$ be the discrete target token from a vocabulary of size $|\mathcal{V}|$. Let $X^m = \{X_i, 1 \leq i \leq m\}$ be a set of $m$ hidden states that are conditionally i.i.d. given $Y$. The training objective is to minimize cross-entropy, which is asymptotically equivalent to minimizing the conditional entropy $H(Y|X^m)$.

**Lemma 2** (Data Efficiency).

$$H(Y|X^m) \geq H(Y) - m \cdot I(Y; X) \tag{5}$$

*Proof.* The goal is to show:

$$H(Y|X^m) \geq H(Y) - mI(X; Y)$$

By the definition of Mutual Information:

$$H(Y|X^m) = H(Y) - I(Y; X^m)$$

We need to bound $I(Y; X^m)$. Apply chain rule of mutual information,

$$I(Y; X^m) = H(X^m) - H(X^m|Y)$$

Since $X_i$ are conditionally independent given $Y$:

$$H(X^m|Y) = \sum_{i=1}^{m} H(X_i|Y)$$

For the first term $H(X^m)$, by sub-additivity of entropy, the entropy of the joint distribution is always less than or equal to the sum of individual entropies (independence maximizes entropy):

$$H(X^m) \leq \sum_{i=1}^{m} H(X_i)$$

Substitute these back into the Mutual Information expansion:

$$I(Y; X^m) \leq \sum_{i=1}^{m} H(X_i) - \sum_{i=1}^{m} H(X_i|Y)$$

$$I(Y; X^m) \leq \sum_{i=1}^{m} \left( H(X_i) - H(X_i|Y) \right)$$

$$I(Y; X^m) \leq \sum_{i=1}^{m} I(Y; X_i)$$

Since $X_i$ are identically distributed, $I(Y; X_i)$ is the same for all $i$:

$$I(Y; X^m) \leq m \cdot I(Y; X)$$

Finally substitute this upper bound on Information back into step 1. Since we are subtracting a larger value, the result is a lower bound on entropy:

$$H(Y|X^m) = H(Y) - I(Y; X^m) \geq H(Y) - m \cdot I(Y; X)$$

$\square$

Suppose $H(Y|X^m) \leq \epsilon$ after training, we have

$$\epsilon \geq H(Y|X^m) \geq H(Y) - m \cdot I(Y; X)$$

Recent theoretical work on infinite-width limits (Yang & Littwin, 2021) establishes that layer pre-activations converge to Gaussian distributions. Motivated by this, we model the local representation dynamics using a canonical Gaussian Channel approximation with additive noise. Specifically, we decompose $X = Z + N$, where $Z$ is the latent signal, and $N \sim \mathcal{N}(0, \sigma^2 I)$ is the additive Gaussian noise. We define the Signal-to-Noise Ratio as

$$\text{SNR} = \frac{\mathbb{E}[\|Z\|^2]}{\mathbb{E}[\|N\|^2]}$$

Under the Gaussian channel approximation, mutual information is a logarithmic function of SNR (Shannon, 1948):

$$I(X;Y) = \frac{1}{2}\log(1 + \text{SNR})$$

Substituting this capacity into Lemma 2, we have

**Corollary 2** (Signal-to-Noise Ratio).

$$m \geq \frac{H(Y) - \epsilon}{\frac{1}{2}\log(1 + \text{SNR})} \tag{6}$$

Corollary 2 indicates that $m$ is inversely proportional to $\log(1 + \text{SNR})$. Consequently, if the Semantic Tube works as expected, it will increase SNR and strictly lower the data requirement $m$.

Let $\hat{Y} = f(X^m)$ be the estimator of $Y$ produced by the model. Let $P_e = P(\hat{Y} \neq Y)$ be the probability of error (incorrect token generation). Fano's Inequality (Cover & Thomas, 1991) provides a lower bound on the conditional entropy $H(Y|X^m)$ in terms of the error probability:

$$H(Y|X^m) \leq H_b(P_e) + P_e \log(|\mathcal{V}| - 1)$$

where $H_b(P_e)$ is the binary entropy function. For LLMs, $|\mathcal{V}| \gg 1$, the term $P_e \log|\mathcal{V}|$ dominates $H_b(P_e)$. Hence we can simplify Fano's inequality to be:

$$H(Y|X^m) \leq P_e \log(|\mathcal{V}| - 1) \tag{7}$$

Plug Equation (5) into Equation (7), immediate we get

**Corollary 3** (Accuracy).

$$P_e \gtrsim \frac{H(Y) - m \cdot \frac{1}{2}\log(1 + \text{SNR})}{\log|\mathcal{V}|} \tag{8}$$

Corollary 3 indicates that if we observe significant improvement on training accuracy, we know that SNR is higher.

## I. Data Efficiency

In this section we present the results of experiments on Llama3 3B and 8B using $\frac{1}{2}$, $\frac{1}{4}$, $\frac{1}{8}$, $\frac{1}{16}$, and $\frac{1}{32}$ of the dataset in Figure 12, where we see similar trend as in Llama3 1B (Figure 1).

## J. Regular Expression Samples

We list in Table 4 a few samples from the SYNTH dataset that end with either ".*" or ".*.*", which are functionally equivalent.

*Table 4.* Regular expression samples from the SYNTH dataset that end with either ".*" or ".*.*", which are functionally equivalent.

| Regular Expressions |
|:---:|
| .*([a-z])\|([AEIOUaeiou])\|([A-Za-z]).* |
| .*([A-Za-z]).*([0-9]).*.* |
| ((dog)(.*)).*([AEIOUaeiou]).* |
| (dog).*((truck)\|([A-Z])\|([0-9])).* |
| .*(.)&([0-9])&(dog).* |
| .*(dog).*((.)*).*.* |
| .*dog.*[a-z].*.* |

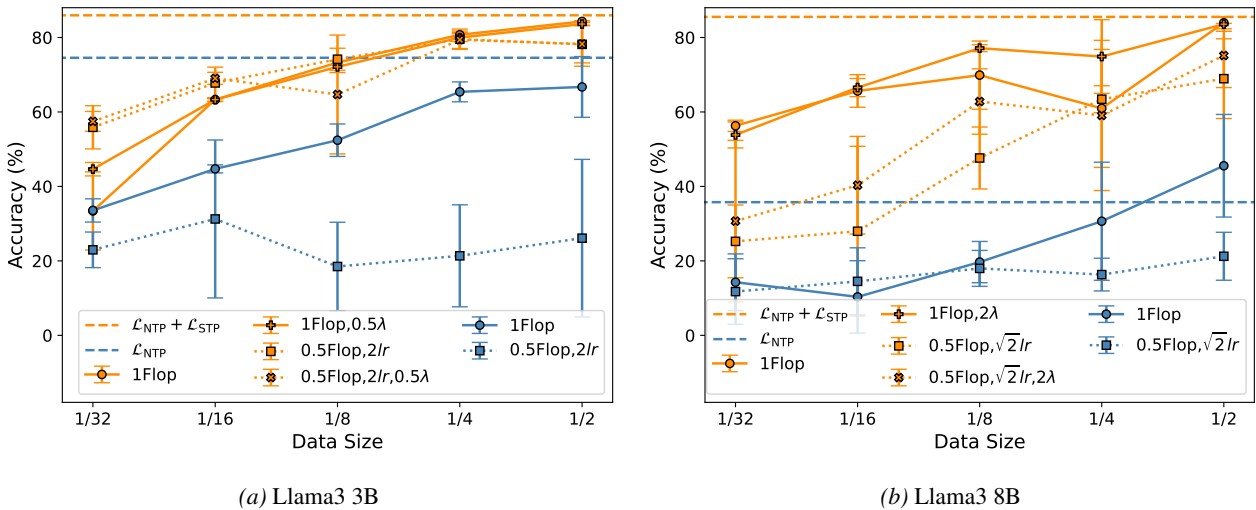

*(a)* Llama3 3B                          *(b)* Llama3 8B

*Figure 12.* Semantic Tube (our approach) and regular fine-tuning with $\frac{1}{2}, \frac{1}{4}, \frac{1}{8}, \frac{1}{16}$, and $\frac{1}{32}$ dataset on (a) Llama3 3B and (b) Llama3 8B.

## K. Tuning $\lambda$

In this section, we present the accuracy vs. $\lambda$ curves for the various configurations of Semantic Tubes, Two Views, and Mask detailed in Section 4.6. As shown in Figure 13, we observe across all cases that the curve is concave, most of the time with a maximum reached at $\lambda$ values between 0.01 and 0.8. Furthermore, when $\lambda$ exceeds the optimal value, we occasionally observe a precipitous drop in accuracy accompanied by a drastic increase in standard deviation. Collectively, these results provide strong evidence supporting the validity of (P4).

## L. Ablation

**Semantic Tube**: We ablate several variations of the Semantic Tube configuration:

- **Zero**: Instead of randomly picking $s$, this variation fixes the start index $s = 0$. The loss becomes $\mathcal{L}_{\text{STP}} = 1 - \cos(h_r - h_0, h_t - h_r)$.

- **Pred**: We introduce a learnable linear projector $P$ and modify the loss to $\mathcal{L}_{\text{STP}} = 1 - \cos(P(h_r - h_s), h_t - h_s)$. aligns the approach more closely with the JEPA style, utilizing a non-identity predictor. $P$ is randomly initialized and trained during fine-tuning.

- **Inst**: We incorporate instructions into the token sequence $x_{\leq t}$. These instructions consist of system prompt such as `"Convert natural language to regular expression"`.

**Two Views**: This configuration adopts the LLM-JEPA style two-view structure, where *query* and *answer* represent two views of the same concept. Note that we retain the $\mathcal{L}_{\text{STP}}$ formulation but fix $s = 0$ and set $r$ to the index of the last token of the *query*.

- **Warmup**: We linearly warm up $\lambda$ throughout the training process.

- **Pred**: Identical to the Pred variation in the Semantic Tube configuration.

- **Mean**: Instead of the difference vector $h_r - h_s$, we use the average embedding $\frac{1}{r-s+1}\sum_{s \leq i \leq r} h_i$. Consequently, the loss becomes $\mathcal{L}_{\text{STP}} = 1 - \cos(\frac{1}{r-s+1}\sum_{s \leq i \leq r} h_i, \frac{1}{t-r+1}\sum_{r \leq j \leq t} h_j)$. This is inspired by BERT Mean Pooling (Kim et al., 2021).

**Mask**: This variation is inspired by BERT mask-and-recover training objective (Devlin et al., 2019). Given a token sequence $x_{\leq t}$, we randomly pick a span $[s, r]$ and replace the tokens within this span with the `[MASK]` token. Let $y_{\leq t}$ denote the

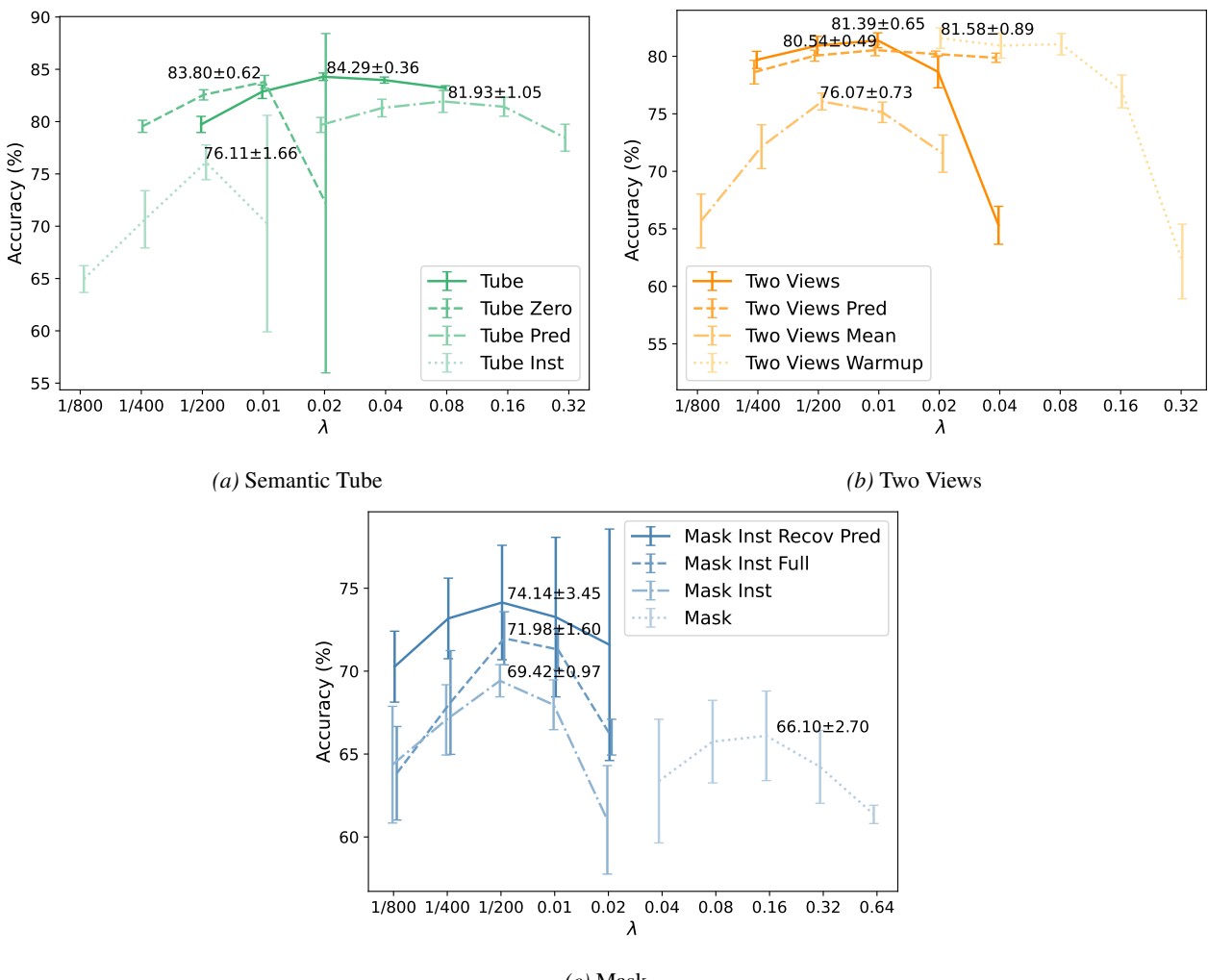

*(a)* Semantic Tube

*(b)* Two Views

*(c)* Mask

*Figure 13.* Tuning $\lambda$ for various configurations of (a) Semantic Tube, (b) Two Views, and (c) Mask. In all cases, the accuracy vs. $\lambda$ curve is concave. We also observe that when $\lambda$ exceeds the optimal value, accuracy declines rapidly while the standard deviation increases sharply, indicating that $\lambda \ll 1$ is preferred.

masked sequence and $g_t = f(y_{\leq t})$. The loss is defined as $\mathcal{L}_{\text{mask}} = 1 - \cos(h_r - h_s, g_t)$. This can be interpreted as recovering the information of the masked tokens using the representation of the masked sequence $y_{\leq t}$.

- **Full**: Instead of aiming to match $h_r - h_s$, we target $h_t$. The loss becomes $\mathcal{L}_{\text{Mask}} = 1 - \cos(h_t, g_t)$, corresponding to the recovery of the full masked sequence rather than just the masked span.

- **Pred**: Identical to the Pred variation in the Semantic Tube configuration.

- **Inst**: Identical to the Inst variation in the Semantic Tube configuration.

**Curvature**: This variation is inspired by the curvature straightening objective (Hénaff et al., 2021). Let $\theta_i$ be the angle between $h_i - h_{i-1}$ and $h_{i+1} - h_i$. The loss is defined as $\mathcal{L}_{\text{Curvature}} = \frac{1}{t} \sum_{i \leq t} |\theta_i|$.

- **Sign**: Replaces $|\theta_i|$ with $\theta_i$ (allowing for signed curvature).

The fact that Pred yields inferior performance in both the Semantic Tube and Two Views configurations supports (P5).

The $p$-values comparing variations and options are presented in Tables 5 and 6.

*Table 5.* Pairwise $p$-values comparing variation families. A cell is populated only if the mean accuracy of the row method exceeds that of the column method. $p$-values are computed using a paired, one-tailed $t$-test, restricted to the best-performing variant from each family.

| | Two View | Mask | Curvature |
|---|---|---|---|
| **LLM-JEPA2** | 1.14e-3 | 1.77e-3 | 3.04e-5 |
| **Two View** | | 4.76e-3 | 5.10e-5 |
| **Mask** | | | 1.28e-4 |

*Table 6.* Pairwise $p$-values comparing options within each variation family. A cell is populated only if the mean accuracy of the row option exceeds that of the column option. $p$-values are computed using a paired, one-tailed $t$-test. Values exceeding 0.05 are struck through.

| | Zero | Pred | Inst | | 2View | Pred | Mean |
|---|---|---|---|---|---|---|---|
| **LLM-JEPA2** | ~~0.0534~~ | 2.03e-3 | 2.34e-4 | **2View+Warmup** | ~~0.265~~ | 0.0426 | 9.16e-6 |
| **+Zero** | | 0.0185 | 5.56e-4 | **2View** | | ~~0.0689~~ | 1.19e-6 |
| **+Pred** | | | 2.97e-4 | **+Pred** | | | 2.78e-4 |

| | Inst,Recov | Inst | Mask | | | Signed |
|---|---|---|---|---|---|---|
| **Mask+*all*** | ~~0.0629~~ | 0.0159 | 4.02e-3 | | | |
| **-Pred** | | 2.34e-3 | 1.18e-3 | **Curvature** | | 0.0368 |
| **-Recov,Pred** | | | 0.0103 | | | |

# M. Pretraining Methodology

For pretraining, we first tune the learning rate for standard pretraining and find $1.6 \times 10^{-4}$ to be optimal. Holding this learning rate fixed, we then tune the STP coefficient $\lambda_{\text{STP}}$ and find that pretraining requires a substantially smaller value than the one used in fine-tuning. This suggests that, when semantic structure has not yet emerged at random initialization, STP should be introduced more gently to avoid disrupting early representation learning.

