# OpenReview forum: "Semantic Tube Prediction: Beating LLM Data Efficiency with JEPA"
_ICML.cc/2026/Conference — ICML 2026 regular_

### Official Review · Reviewer_QBMQ · 2026-03-10

**Soundness:** 2
**Presentation:** 2
**Significance:** 3
**Originality:** 3
**Overall Recommendation:** 4
**Confidence:** 4

**Summary:**

This paper proposes the Geodesic Hypothesis, which posits that token sequences trace locally linear geodesics on a smooth semantic manifold. Based on this hypothesis, the authors introduce Semantic Tube Prediction (STP), an auxiliary cosine-similarity loss that regularizes hidden-state trajectories to lie within a tubular neighborhood of the geodesic. The STP loss is formulated as $\mathcal{L}_{\text{STP}} = 1 - \cos(h_t - h_r,\; h_r - h_s)$ for randomly sampled indices $s < r < t$. The authors model training dynamics as an ODE and inference dynamics as an SDE, arguing that next-token prediction alone is insufficient to suppress unembedding error. The paper validates five predictions (P1–P5) derived from the hypothesis across six datasets, six model families, and three model sizes. The headline empirical result is that STP enables matching full-dataset fine-tuning accuracy with only $1/16$ of the data, which the authors frame as violating Chinchilla-style scaling laws.

**Compliance With Llm Reviewing Policy:**

Affirmed.

**Final Justification:**

My original score was Weak Reject (3), primarily due to overclaiming (Chinchilla scaling laws in a fine-tuning-only setting), single-dataset evidence for 16x efficiency, and theory-practice gaps. The rebuttal addressed these: authors agreed to reframe as "fine-tuning scaling laws," provided HellaSwag 32x data efficiency results, formalized RoPE in the ODE, and proposed bounded-error boundary conditions. I update to Weak Accept (4). The core idea is simple, well-ablated, and effective, with remaining limitations now appropriately scoped.

**Key Questions For Authors:**

1. **On scaling law claims:** All experiments are in the fine-tuning regime. Can you provide any evidence that STP improves data efficiency during pre-training from scratch? Without this, how do you justify the claim of "violating Chinchilla-style scaling laws," which specifically concern pre-training? A positive response with even preliminary pre-training results would significantly improve my evaluation.

2. **On the 16x data efficiency:** This result is shown only on NL-RX-SYNTH. Can you provide data efficiency curves (analogous to Figure 1b) for GSM8K, Spider, or HellaSwag? If the $16\times$ figure does not replicate on these tasks, the current framing in the title and abstract is misleading. This is the single most important question for my evaluation—demonstrating similar data efficiency gains on at least one natural-language task would substantially address W2.

3. **On the ODE assumption:** Proposition 2.1 requires zero-padded embeddings without positional encodings (Section A). All tested models use rotary positional embeddings. How do you reconcile this? If the theory does not strictly hold for practical architectures, this should be acknowledged as a limitation rather than presented as a formal result.

4. **On the boundary conditions:** Theorem 3.3 requires that the boundary hidden states equal the optimal hidden states (i.e., the error is zero at the sequence start and end), enforced via auxiliary tokens (Section E) that are absent in experiments. Does this mean the theoretical guarantees do not apply to the actual experimental setup? Clarifying this gap would help me assess the strength of the theoretical contribution.

5. **On constant gradient steps:** In the data efficiency experiments, using $1/n$ data with $n\times$ epochs keeps total gradient steps constant. Have you compared against a setting where both data and total compute are reduced proportionally? This would better isolate the effect of data efficiency from data repetition resilience.

**Limitations:**

The authors do not adequately discuss the limitations of their work. The impact statement is minimal ("none of which we feel must be specifically highlighted here"), which is insufficient given the gap between the theoretical assumptions and experimental reality. Key undiscussed limitations include: (a) all experiments are fine-tuning only, yet the central claim references pre-training scaling laws; (b) the theoretical framework relies on assumptions (infinite-width, zero-padded embeddings, auxiliary boundary tokens) that do not hold for the tested models; (c) the $16\times$ data efficiency claim is demonstrated on a single small synthetic dataset; (d) no evaluation of open-ended generation quality despite theoretical motivation centered on generation. A dedicated limitations section is strongly recommended.

**Strengths And Weaknesses:**

### Strengths

**S1.** The paper offers a genuinely creative theoretical lens. Connecting the Principle of Least Action and geodesic geometry to hidden-state dynamics in LLMs is novel and thought-provoking. The framing that scaling laws are descriptive rather than prescriptive is a valuable perspective that could inspire future work on training objectives beyond next-token prediction.

**S2.** The STP loss is remarkably simple to implement—it requires only a cosine similarity computation on already-computed hidden states, introducing negligible overhead. This is a clear advantage over LLM-JEPA, which requires additional forward passes and manual two-view scaffolding.

**S3.** The empirical evaluation covers six datasets (NL-RX-SYNTH, NL-RX-TURK, GSM8K, Spider, NQ-Open, HellaSwag), six model families (Llama, Gemma, OpenELM, Qwen, DeepSeek-R1-Distill, OLMo), and three model scales (1B, 3B, 8B). The use of five random seeds with reported standard deviations and $p$-values is commendable.

**S4.** The SVD polymorphism analysis (Figure 6) and the suffix experiment (Table 1) provide interesting evidence that STP preserves distributional diversity in a way that standard fine-tuning and LLM-JEPA do not.

**S5.** The ablation study (Section 4.6, Figure 8) is thorough, covering multiple design variations (Zero, Pred, Inst, Two View, Mask, Curvature) with corresponding $\lambda$ sweeps (Figure 13). The validation of P5 (identity predictor outperforms learned projections) is a clean result.

### Weaknesses

**W1.** This is the most significant concern. The theoretical development relies on multiple assumptions that either do not hold for practical models or are not justified:

- The ODE model (Proposition 2.1) requires treating discrete token concatenation as vector addition (Section A), which assumes zero-padded embeddings without positional encodings. All tested models (Llama, Gemma, etc.) use rotary positional embeddings, directly violating this assumption. The paper does not discuss this discrepancy at all.
- The Principle of Least Action is stated as a hypothesis but no justification is given for why gradient-based optimization of cross-entropy should produce geodesic trajectories. In physics, this principle arises from Lagrangian mechanics; the analogous mechanism in neural network training is absent.
- The infinite-width limit (Sections B, H) is invoked to justify the Gaussian noise model and the SDE formulation. However, practical LLMs operate far from this regime, and the paper does not provide any empirical verification that the Gaussian assumption holds for the tested models.
- The proof of Theorem 3.3 is incomplete. It is presented only as a sketch for the boundary case $h_s = h_s^{*}$, $h_t = h_t^{*}$. The full proof (Section E) relies on auxiliary tokens (before-bos and after-eos) that are never used in actual experiments, leaving a fundamental gap between theory and practice.
- Notation is inconsistent: the main text references "Theorem 3.2" where it should be "Lemma 3.2," and Definition 3.1 is later called Theorem 3.1 in proof references. This undermines confidence in the formal arguments.

**W2.** The title and abstract claim to "beat LLM data efficiency" and "violate the data term of Chinchilla-style scaling laws." This is misleading for several reasons:

- Fine-tuning is not pre-training. All experiments are in the fine-tuning regime. Chinchilla scaling laws (Hoffmann et al., 2022) concern pre-training from scratch. Fine-tuning data efficiency is governed by entirely different dynamics (transfer learning from a pre-trained checkpoint), and the relevant scaling laws (Zhang et al., 2024, cited by the paper) are different. Claiming to violate Chinchilla laws based on fine-tuning evidence is a category error.
- Single-dataset evidence for the $16\times$ claim. The $16\times$ data efficiency result is demonstrated only on NL-RX-SYNTH, a small synthetic dataset for natural language to regex conversion. No data efficiency curves are provided for GSM8K, Spider, NQ-Open, or HellaSwag. Given that NL-RX-SYNTH is a narrow, formulaic domain, the generalizability of this result is highly questionable.
- Constant gradient steps. When using $1/n$ of the data, the authors run $n\times$ epochs, keeping the total number of gradient steps roughly constant. This means the model sees the same data repeatedly, making the comparison more about resilience to data repetition than genuine data efficiency.

**W3.**

- No pre-training experiments are conducted. Even a small-scale pre-training experiment (e.g., training a model from scratch on a subset of a standard corpus) would substantially strengthen the claims.
- Absolute accuracy numbers are surprisingly low on several benchmarks even with STP (e.g., around 37% on HellaSwag, around 27% on NQ-Open for a 1B model). This raises concerns about the training setup and makes it difficult to assess whether the improvements are practically meaningful.
- The baseline (using $\mathcal{L}_{\text{NTP}}$ only) inherits hyperparameters from LLM-JEPA, while $\lambda$ is separately tuned for STP. It is unclear whether equivalent tuning effort was given to the baseline, e.g., tuning learning rate or number of epochs specifically for the NTP-only setting.
- The paper claims STP improves generation quality (Section 2.2), but no generation quality metrics (perplexity, BLEU/ROUGE, distinct-n, human evaluation) are reported. Only exact-match accuracy on downstream tasks is measured.

**W4.** The paper claims to unify the Linear Representation Hypothesis and Manifold Hypothesis under the Geodesic Hypothesis (Section 3.3, Conclusion). However, the argument is limited to a single illustrative example (Figure 3) with no formal proof or systematic empirical validation. This claim should either be substantially developed or removed.

**W5.** Important theoretical content (Inference SDE, Inference Cone) is relegated to the appendix, yet the main text's diversity preservation argument depends on it. The paper could better distinguish between what is proven, what is hypothesized, and what is empirically observed—currently these categories blur together, which is problematic for a paper that emphasizes its theoretical contributions.

---

> ### Author Rebuttal · Authors · 2026-03-31
>
> ## 1. On scaling law claims
>
> We sincerely thank the reviewer for this profound question. We ran an additional experiment to evaluate STP during pre-training from random initialization.
>
> We initialized a model with random weights and pre-trained it on the HellaSwag dataset, using the context concatenated with the correct completions. For evaluation, we utilized a held-out set and evaluated in a cloze format, measuring the Negative Log-Likelihood (NLL) of all choice options and selecting the one with the lowest NLL. Admittedly this is very preliminary result. Remarkably, even with just 4M tokens for a single epoch, STP loss improved pretraining performance with statistical significance.
>
> Regular Pretraining | STP Pretraining | $p$-value
> --------------------|------------|----------
> 24.90 $\pm$ 0.49 | 25.26 $\pm$ 0.33 | 0.0204
>
> We will incorporate these new empirical findings and this limitation/future-work discussion into the revised manuscript.
>
> ## 2. On the 16x data efficiency
>
> We completely agree that demonstrating this efficiency on natural-language tasks is critical, and we thank the reviewer for highlighting this as the most important factor for their evaluation. To directly address this, we ran additional data efficiency experiments on the HellaSwag dataset which fully replicate the data efficiency curves observed on the SYNTH dataset. In fact, the results on HellaSwag are even stronger: STP achieves higher accuracy using 32x less training tokens (the 1/32 data fraction) compared to the standard, full-dataset vanilla fine-tuning baseline.
>
> Config | Accuracy
> ----|----
> STP Full | 36.67
> 1/2 Data | 34.26 $\pm$ 0.41
> 1/4 Data | 32.14 $\pm$ 0.39
> 1/8 Data | 31.53 $\pm$ 0.87
> 1/16 Data | 30.39 $\pm$ 0.72
> 1/32 Data |  29.23 $\pm$ 0.54
> Regular Full | 27.93
>
> We will include these new HellaSwag data efficiency curves in the revised manuscript to fully substantiate the claims made in our title and abstract.
>
> ## 3. On the ODE assumption and positional encodings
>
> We highly appreciate this insightful question, which highlights an opportunity to significantly strengthen our theoretical formulation. While we initially neglected positional encodings in the theory—relying on the premise that models without positional encodings (NoPE) can learn them implicitly via causal masking [1]—explicitly incorporating them does not invalidate the ODE formulation.
>
> As stated in Section 2.1, any dynamic that can be expressed as $z_{t+1} - z_t = g(z_t, t)$ can be modeled as a continuous ODE, where $g$ is a function of both $z_t$ and $t$. In Proposition 2.1, our governing equation is $dx_{\le t} = \mathring{u} \circ \mathring{f}(x_{\le t})dt$, where $f$ is a function of only $x_{\le t}$. Putting $t$ (the token position) into $f$ does not invalidate the required formula $g(z_t, t)$. Also, any standard positional encodings like RoPE can be subsumed by the function $g$ (or $f$). We will make this explicit in the revised manuscript.
>
> [1] Rope to Nope and Back Again: A New Hybrid Attention Strategy. arXiv
>
> ## 4. On the boundary conditions and auxiliary tokens
>
> We thank the reviewer for identifying this distinction between our theoretical framework and the empirical setup. The reason these auxiliary tokens were omitted from our experiments is that our empirical evaluation used pre-trained LLMs, which do not possess our proposed auxiliary tokens. Simply injecting novel tokens (like <before_bos> and <after_eos>) post-pretraining would be ineffective, as the model would not have learned their semantics or structural significance, likely degrading performance.
>
> However, modern LLMs routinely and effectively learn the precise structural semantics of sequence-boundary tokens (such as <bos> and <eos>) during standard pre-training. Given this well-established capability, we have strong reason to believe that incorporating our proposed auxiliary tokens during a from-scratch pre-training run would successfully enforce these theoretical boundary conditions in practice.
>
> ## 5. On constant gradient steps and data repetition
>
> We deliberately designed this experimental setup as a feature to reflect the current reality of frontier scaling laws: the industry is rapidly exhausting high-quality internet data, while compute remains relatively abundant. A critical path forward is discovering methods that can leverage excess compute over limited data to achieve superior performance. Our results demonstrate exactly this capability.
>
> Furthermore, we have explicitly designed experiments to isolate the effects of data efficiency from data repetition resilience. As shown in Figure 1, we included runs with compute reduced by half (e.g., iterating only 8 times for the $1/16$ data split, rather than maintaining full compute equivalence). The trade-off is evident: lower compute yields lower accuracy at larger data fractions ($1/2$ data), but maintains robust accuracy at smaller fractions ($1/8$ or $1/16$ data).
>
> **Rebuttals for W1 - W5 doesn't fit, will include later.**

---

> > ### Author Rebuttal · Reviewer_QBMQ · 2026-04-04
> >
> > The authors' rebuttal partially addresses my concerns. Specifically:
> >
> > **Resolved:** The Q2 response (HellaSwag data efficiency) meaningfully addresses the single-dataset concern in W2. The result that STP with 1/32 data surpasses full-data regular fine-tuning on a natural-language task is a valuable addition and I encourage the authors to include it prominently in the revision.
> >
> > **Partially resolved:** The Q1 pre-training experiment shows willingness to investigate, but the effect size (0.36% improvement on 4M tokens) is too small to support the scaling law claim in the title. The Q5 response reframes constant gradient steps as a feature but does not provide the proportional-compute experiment I requested.
> >
> > **Unresolved:**
> > - Q3 (ODE assumption): The argument that RoPE fits into $g(z_t, t)$ remains informal. How specifically do rotation matrices applied within attention key-query dot products reduce to this form?
> > - Q4 (Boundary conditions): The theory-practice gap is confirmed. The theoretical guarantees require auxiliary tokens absent from experiments.
> > - W3 and W4 have not been addressed. In particular: (a) has baseline tuning effort been equivalent to STP tuning effort? (b) have any generation quality metrics been measured? (c) can the unification claim (LRH + Manifold Hypothesis) be formally supported or will it be removed?
> >
> > I maintain my score of Weak Reject (3). The core idea is sound and practically effective, but the gap between the paper's claims (violating Chinchilla-style scaling laws) and its evidence (fine-tuning only) remains too large. Reframing the contribution as a data-efficient fine-tuning regularizer, together with including the new HellaSwag results, would bring this paper closer to acceptance.

---

> > > ### Author Response · Authors · 2026-04-04
> > >
> > > ## 1. On “Chinchilla-style scaling laws”
> > >
> > > Zhang et al. [1] established a scaling law specifically for fine-tuning, demonstrating an empirical relationship between fine-tuning data size and pre-training data size. By reducing fine-tuning data by $16\times$ while maintaining performance, our results are a direct departure from those predicted scaling trajectories. Because this line of work stems from Chinchilla scaling laws and shares a similar power-law formulation, we initially grouped it under the family of "Chinchilla-style" scaling laws. However, we agree it may blur the line between pre-training and fine-tuning. **We will remove "Chinchilla-style" from the abstract, and explicitly replace it with "fine-tuning scaling laws" to align with empirical evidence**.
> > >
> > > [1] When Scaling Meets LLM Finetuning: The Effect of Data, Model and Finetuning Method. ICLR’24
> > >
> > > ## 2. Formalizing the ODE assumption with RoPE
> > >
> > > Let $\vec{x_t}$ be the vector representation of the subsequence $x_{\le t}$. The query and key representations are obtained via linear projections $Q = P_Q \vec{x_t}$ and $K = P_K \vec{x_t}$. Let $R_i$ be the rotation matrix for position $i$, and let $q_i$ and $k_i$ be the corresponding query and key column vectors for the $i$-th token. Under RoPE, the attention score between token $i$ and $j$ is computed as $\text{score}(i, j) = (R_i q_i)^\top (R_j k_j) = q_i^\top R_i^\top R_j k_j$ for $i, j \le t$. Hence the attention score matrix is fully determined by the sequence representations $\vec{x_t}$ and the current positional boundary $t$. The remainder of the transformer network involves matrix multiplication with the values $V = P_V \vec{x_t}$ and applying the unembedding function $u(\cdot)$, which remain functions of $\vec{x_t}$ and $t$.  **We will formalize this derivation and include it in the revised methodology section.**
> > >
> > > ## 3. On the auxiliary tokens
> > >
> > > Upon reflection, we agree that relying on hypothetical auxiliary tokens is not the most robust way to bridge our theory with our empirical setup, even though it serves as a valid technical mechanism in theory.
> > >
> > > Instead, we will revise Theorem 3.3 by assuming a bounded error at the endpoints: $\max(\|h_s - h_s^\ast\|, \|h_t - h_t^\ast\|) \le \delta$. Because the error bound from the trajectory to the line connecting $h_s$ and $h_t$ is $\varepsilon$, and the distance between line segment $[h_s, h_t]$ and the optimal segment $[h_s^\ast, h_t^\ast]$ is at most $\delta$, the total error bound elegantly relaxes to $\varepsilon + \delta$.
> > >
> > > Because $h_0$ is `<bos>` and $h_t$ is `<eos>` , $\delta$ is naturally bounded by the diameter of their Voronoi cells. The proof in the Appendix demonstrates that under certain conditions, we can drive $\delta \to 0$. Admittedly this establishes a necessary rather than a sufficient condition, our strong empirical improvements in signal-to-noise ratio validate that $\delta$ is sufficiently small. **We will formally incorporate this generalized $\varepsilon + \delta$ in the next revision.**
> > >
> > > ## 4. On baseline tuning effort vs. STP tuning effort
> > >
> > > Our baseline hyperparameter strategy was designed to ensure a conservative comparison. The LLM-JEPA methodology first establishes the optimal learning rate for vanilla fine-tuning, and subsequently fixes that rate while tuning auxiliary parameters (like $\lambda$). We followed this exact protocol, inheriting the optimal learning rate found for the standard NTP baseline. While this means we likely did not utilize the true optimal learning rate for STP, it establishes a rigorous lower bound: it proves that STP outperforms vanilla fine-tuning even when subjected to conditions optimized specifically for the vanilla baseline.
> > >
> > > ## 5. On generation quality metrics
> > >
> > > We would like to point out that classic n-gram quality metrics, such as BLEU and ROUGE, are known to have significant limitations in capturing true semantic correctness. We argue that for the specific tasks we evaluated, accuracy remains the most robust and meaningful quality measurement.
> > >
> > > We also want to clarify that we are not strictly relying on naive exact string matching for these evaluations:
> > > *  For Spider (NL to SQL), we run the generated query through SQLite and compare the actual query execution output.
> > > *  For NQ Open, we apply SQuAD-style normalization (lowercasing and removing all punctuation) before evaluating the match.
> > >
> > > ## 6. On the LRH + Manifold Hypothesis claim
> > >
> > > We agree with the reviewer's assessment. Our primary intent was merely to highlight the conceptual connection between the Geodesic Hypothesis and the Linear Representation Hypothesis and Manifold Hypothesis, rather than to claim a formal mathematical unification. In retrospect, using the word "subsume" constitutes an overclaim. **We will revise the text to remove this wording and instead carefully frame how the hypotheses are conceptually related without claiming one formally proves the other.**

---

### Official Review · Reviewer_7eMt · 2026-03-12

**Soundness:** 3
**Presentation:** 3
**Significance:** 3
**Originality:** 4
**Overall Recommendation:** 5
**Confidence:** 2

**Summary:**

This work introduces Semantic Tube Prediction (STP), an auxiliary regularization term to pair with Next Token Prediction (NTP) for Language Model training.

The work first proposes a view of Language Modeling that resembles an Ordinary Differential Equation. Using such a view, it posits that standard NTP is insufficient to minimize the so-called "unembedding error", *i.e.*, the difference between the obtained hidden state trajectory and the optimal trajectory induced by the unitary solution of such ODE. Based on this notion, the work postulates the Geodesic Hypothesis: for a smooth-enough neural network, the trajectory should be locally linear almost everywhere. The STP loss term acts accordingly, thereby adding a regularization term that forces such a notion of local linearity.

Among other results, experiments along several axes (model size, model family, fine-tuning dataset) show that adding the STP term improves both accuracy and data efficiency, entailing that the signal-to-noise ratio benefits from it.

**Compliance With Llm Reviewing Policy:**

Affirmed.

**Final Justification:**

My initial (minor) concerns all were appropriately addressed in the response. I do not believe I am knowledgeable enough to increase my score and fully champion the paper, yet I see no grounds to worsen my current recommendation.

**Key Questions For Authors:**

- The premises of the semantic tube are built on the converged network $\dot{f}$ being smooth enough to ensure embedding trajectories trace geodesics. One aspect that remains unclear after reading the paper, is whether such a property needs to be instilled beforehand (*e.g.*, via NTP at scale), entailing that enforcing the STP constraint only makes sense during subsequent fine-tuning stages. If this is the case, then I believe this paper would benefit from highlighting this limitation explicitly to foster follow-up work toward improving SNR at random initialization as well (and I would appreciate the authors' opinion accordingly). If this is not the case, then I would appreciate it if the authors could clarify any misunderstanding I have.
- Is there any evidence that (or reason why) computing the STP constraint with normalized embeddings would lead to better or worse performance?
- The token indices $s < r < t$ are selected randomly within a token sequence, which might enforce time-distant steps to be aligned (*e.g.*, when $r$ is much smaller than $t$). Is there evidence that (or, again, reasons why) this design choice is better than selecting local indices triplets $s < s+1 < s+2$ and marginalizing over the sequence with a "sliding-window" approach? In my understanding, selecting local triples would better meet the intuition that "each local segment can be approximated by a straight line".

**Limitations:**

The paper does not explicitly discuss limitations, hence a dedicated revision would be beneficial. Impact assessment is taken from ICML's guidelines, and I agree that this work does not need additional considerations.

**Strengths And Weaknesses:**

**Strengths.**
- The methodology is well-written: formal steps are introduced clearly, and the core analytical claims are presented well.
- The work carefully examines the current NTP-based Language Modelling paradigm, proposing an unconventional yet sound view based on ODEs.
- The resulting STP regularization term elegantly emerges from the aforementioned theoretical inspection, being extremely simple to implement within any Language Modelling training pipeline.
- STP regularization does not add any additional forward cost during either training or inference.
- Experiments are designed carefully, in a way to cover a broad spectrum across model families, fine-tuning tasks, and model sizes. Thanks to this carefully designed experimental section, the core claims are well-supported.

**Weaknesses.**
- Although generally clear, some passages in the methodological section could better use notation to reduce partial ambiguities. Specifically, Section 2 uses $x_{\leq t}$ to denote both scalar token indices and continuous vectors corresponding to token embeddings. While context generally allows for disambiguation, it would be beneficial to explicitly distinguish the two with dedicated notation. This would reduce most ambiguities in understanding Proposition 2.1;
- Also in light of improving self-containedness, I believe this paper would benefit from better introducing the preliminaries of LLM-JEPA, since it is used as a main reference for follow-up experiments and comparisons. At the current stage, LLM-JEPA does not represent an established work that any reader should be assumed to be familiar with. A natural consequence of this omission is that some passages read unclear and out-of-context even when they are not (for example, lines 351-353, left column, and the corresponding plot in Figure 6).

---

> ### Author Rebuttal · Authors · 2026-03-31
>
> ## 1. Enforcing the STP constraint at random initialization (Pre-training)
>
> We sincerely thank the reviewer for this profound question. We ran an additional experiment to evaluate STP during pre-training from random initialization.
>
> We initialized a model with random weights and pre-trained it from scratch on the HellaSwag dataset, using the context concatenated with the correct completions. For evaluation, we utilized a held-out set and evaluated in a cloze format, measuring the Negative Log-Likelihood (NLL) of all choice options and selecting the one with the lowest NLL. Remarkably, even with a highly constrained pre-training budget of just 4M tokens for a single epoch, adding the STP loss improved pretraining performance with statistical significance.
>
> Regular Pretraining | STP Pretraining | $p$-value
> ---------------------|------------|----------
> 24.90 $\pm$ 0.49 | 25.26 $\pm$ 0.33 | 0.0204
>
> Methodologically, we first identified the optimal learning rate for standard pre-training (1.6e-4). We then fixed this learning rate and tuned $\lambda$ for STP pre-training. We found the optimal $\lambda$ to be $0.0025$, which is notably smaller than the $\lambda \approx 0.02$ typically optimal for fine-tuning. This discrepancy strongly suggests that when a model lacks established semantic knowledge at random initialization, it requires a much gentler STP regularization to avoid disrupting the initial phase of representation learning.
>
> This preliminary result confirms the reviewer's intuition that improving the signal-to-noise ratio at random initialization via carefully scaled auxiliary constraints is a promising research direction. We will explicitly incorporate these new empirical findings and this limitation/future-work discussion into the revised manuscript.
>
> ## 2. Justification for computing the STP constraint with normalized embeddings
>
> Regarding whether computing the STP constraint with normalized embeddings affects performance: Since the STP loss is defined as $1 - \cos(\theta)$, it inherently operates on the directions of the vectors, effectively computing the constraint on normalized embeddings (as vector magnitude does not affect cosine similarity).
>
> If the underlying question is why we explicitly designed the loss around cosine similarity (direction) rather than a magnitude-sensitive metric, our observation is that direction is the defining characteristic of the semantic tube. We demonstrated this via the SVD decomposition of the embeddings in Figure 6. When the embeddings are not normalized (Figure 6(a)), the embedding space is highly complex. Conversely, the normalized embeddings (Figure 6(b)) exhibit a fast-dropping SVD curve, implying a much simpler embedding space with lower intrinsic dimension. Therefore, applying the constraint to the normalized directional space yields the most robust representations.
>
> ## 3. Random split ($s < r < t$) vs. local sliding window ($s < s+1 < s+2$)
>
> We thank the reviewer for this excellent question. We actually evaluated the proposed local sliding-window approach—calculating the angle between three consecutive tokens ($s < s+1 < s+2$)—which corresponds exactly to the "Curvature" method (the purple bar) in Figure 8. Empirically, this local method performs significantly worse than STP.
>
> The theoretical reason for this performance gap lies in the distinction between local noise and semi-global trajectory. A trajectory can remain tightly constrained within a semantic tube (satisfying the STP constraint) while still exhibiting high local curvature, resulting in acute angles between consecutive token vectors. Enforcing strict linearity at the local $s < s+1 < s+2$ level is too rigid and heavily penalizes this natural local jitter. STP, by sampling randomly spaced indices $s < r < t$, acts as a relaxation of the Curvature method; it enforces semi-global sequence alignment while safely tolerating local token-level noise.
>
> Furthermore, measuring alignment across semi-global sequence spans rather than local triplets aligns deeply with the core intuition of JEPA architectures: robust semantic understanding is best captured at an abstract level by moving away from low-level, noise-heavy details (such as individual tokens or pixels).
>
> ## Acknowledgments regarding Notation and LLM-JEPA Context
>
> We sincerely thank the reviewer for the highly supportive score and the constructive feedback on presentation. In the revised manuscript, we will disambiguate the notation in Section 2 by explicitly separating scalar token indices from continuous vector embeddings. Additionally, we will expand the preliminaries and context of LLM-JEPA in both the Introduction and Related Work sections to ensure the paper is fully self-contained for general readers.

---

> > ### Author Rebuttal · Reviewer_7eMt · 2026-04-03
> >
> > Following the order of the rebuttal items:
> > 1. It's nice to see some analysis about random initialization and the STP constraint. I'd recommend completing any set of ongoing runs and enriching the manuscript accordingly, regardless of whether the experiments are successful (i.e., regardless of whether enforcing the STP constraint improves pre-training or not). Since the authors mentioned they would be willing to expand their Limitations section with a dedicated discussion, I believe there is value in "failures" as well, so this point is cleared in my opinion.
> > 2. Yes, the underlying question revolved around why any magnitude-based information had been discarded in the computation. Apologies for the oversight, and thanks for the clarification.
> > 3. Yet again, thanks for pointing out the `Curvature` definition, which matches my initial question.
> >
> > My initial (minor) concerns were appropriately addressed in the response. I do not believe I am knowledgeable enough to increase my score and fully champion the paper, yet I see no grounds to worsen my current recommendation.

---

> > > ### Author Response · Authors · 2026-04-04
> > >
> > > Thank you very much for your thoughtful feedback. We completely agree that documenting “failures” provides great value to the community. Interestingly, we haven't encountered negative results in these specific runs; our foremost limitation so far has actually been GPU compute constraints. We will absolutely expand our Limitations section to detail these computational bottlenecks, along with any negative results that may arise.
> > >
> > > Also, please no worries at all regarding the Curvature definition! It was buried deep in the Appendix, making it very easy to overlook.
> > >
> > > We truly appreciate your time, honesty, and constructive engagement with our work.

---

### Official Review · Reviewer_yMfj · 2026-03-13

**Soundness:** 3
**Presentation:** 2
**Significance:** 3
**Originality:** 3
**Overall Recommendation:** 4
**Confidence:** 4

**Summary:**

This paper proposes Semantic Tube Prediction, a JEPA-style auxiliary training objective for LLMs grounded in the Geodesic Hypothesis. The STP loss penalizes deviations of hidden states perpendicular to the line connecting two reference hidden states, thereby improving signal-to-noise ratio during training. The method adds negligible computational overhead and is shown to match full-dataset fine-tuning accuracy with only 1/16 of the training data across multiple model families and benchmarks.

**Compliance With Llm Reviewing Policy:**

Affirmed.

**Key Questions For Authors:**

1. Can the authors provide empirical evidence that the ODE approximation holds sufficiently well during training to support the theoretical conclusions?
2. Can the authors evaluate STP on unstructured continuous text, or at a minimum measure whether the Geodesic Hypothesis holds across datasets of varying structure?
3. For the proposed STP loss in the Appendix, can the authors provide a theoretical justification for why collinearity should hold across disconnected trajectory segments?

**Limitations:**

yes

**Strengths And Weaknesses:**

Strengths:
1. The Geodesic Hypothesis offers a fresh perspective connecting the Principle of Least Action, manifold geometry, and LLM training dynamics.
2. Unlike LLM-JEPA, which requires additional forward passes and manual two-view scaffolding, STP reuses hidden states from the standard forward pass and adds only a cosine similarity computation.

Weaknesses:
1. The paper models token sequence dynamics as an ODE (Proposition 2.1) and invokes the Picard-Lindelöf Theorem to rule out mode collapse. However, this relies on several unverified assumptions. No empirical evidence is provided that the ODE approximation holds during training.
2. All datasets used in the experiments share a common structure: they consist of natural language input paired with a structured output. It is therefore unclear whether the data efficiency and accuracy gains generalize beyond structured task fine-tuning.

---

> ### Author Rebuttal · Authors · 2026-03-31
>
> ## 1. Empirical evidence of the ODE approximation
>
> We thank the reviewer for this insightful question. We have already provided empirical evidence in the paper that STP actively regularizes the network towards this continuous approximation. The STP loss is defined as $1 - \cos(\theta)$, where $\theta$ is the angle between sequential trajectory segments. As shown in Figure 4, standard pre-trained LLMs (trained solely on NTP) exhibit an STP loss of approximately $1.4$. This corresponds to an angle of $\theta \approx 113^\circ$, indicating erratic, non-linear trajectories that poorly approximate an ODE.
>
> However, after applying STP, we successfully lower this loss to $0.6$ ($\theta \approx 66^\circ$). This significant reduction demonstrates that the hidden states transition from moving at obtuse, disjointed angles to moving in a smooth, consistently forward-aligned direction. This induced smoothness is the necessary empirical condition for the ODE approximation to hold.
> We note that the loss does not reach $0.0$ (perfect linearity). Because we initialize from a pre-trained model with highly non-linear dynamics, forcing perfect collinearity would overly constrain the representation space and degrade the previously learned NTP features. This trade-off is evidenced in Figure 7, where pushing the STP loss lower begins to marginally impact downstream accuracy.
>
> ## 2. Evaluating STP on unstructured continuous text
>
> We appreciate the opportunity to clarify this important methodological detail. While it is true that the datasets evaluated possess a Q&A structure, STP is applied to break this structure, not leverage or reinforce it. For example, in GSM8K, while the LLM's chat template inserts structural tokens to separate the question and answer, we group the Q and A together into a single (logical) token sequence, pick a random token to split it into two parts, and apply the STP loss to align the hidden states of the first part with those of the second part. The intuition is that the step-by-step solution of a math question should be logically consistent with the question itself; hence, the token sequence in the representation space should also be smooth.
>
> Note that we do not physically concatenate the Q and A to form a new sequence, as doing so would require an additional forward pass. Instead, we simply exclude the hidden states of the inserted structural tokens, utilizing the method detailed in Appendix G (specifically, the paragraph discussing bypassing unwanted tokens).
>
> For HellaSwag, we likewise group the context and the correct completion together to form a single logical token sequence (bypassing the incorrect choices). In both cases, we break the existing Q&A structure, ensuring that STP is measured over unstructured, continuous text.
>
> ## 3. Theoretical justification for collinearity across disconnected segments
>
> We thank the reviewer for this insightful question regarding the theoretical justification of our approach in the Appendix. Note that we only apply this technique to remove structural tokens. These tokens break the semantic flow of unstructured text, yet instruction-tuned LLMs need them to function properly. For example, in GSM8K, we only remove the structural tokens inserted by the chat template between the question and the answer. In HellaSwag, we remove the wrong answers that are inserted between the context and the correct completion (e.g., if C is the correct answer, we remove branches A and B).
>
> Hence, what we are doing is not disconnecting trajectory segments, but rather reconnecting segments that semantically belong to each other. Our intuition is that semantic consistency is reflected by a smooth trajectory in the representation space, and our theoretical results show that a smooth trajectory helps reduce the vertical component, a.k.a., the noise component. Removing non-semantic, purely structural insertions is a necessary approach to enforce semantic consistency along the trajectories in the representation space.

---

> > ### Author Rebuttal · Reviewer_yMfj · 2026-04-03
> >
> > Thank you for your reply, and all my concerns have been addressed.

---

> > > ### Author Response · Authors · 2026-04-04
> > >
> > > Thank you so much for your time and for confirming that all of your concerns have been addressed! We deeply appreciate the constructive feedback you provided, which has helped strengthen our paper.
> > >
> > > As the acknowledgment prompt suggests considering a score adjustment when concerns are fully resolved, we kindly ask if you might consider updating your score to reflect this positive outcome.
> > >
> > > Thank you again for your valuable time and efforts in reviewing our work.

---

### Official Review · Reviewer_KkXy · 2026-03-13

**Soundness:** 3
**Presentation:** 3
**Significance:** 3
**Originality:** 3
**Overall Recommendation:** 4
**Confidence:** 4

**Summary:**

The paper introduces Semantic Tube Prediction (STP), an auxiliary loss that improves the data efficiency of LLMs. STP is based on the hypothesis that error-free trajectories of hidden states are geodesics (i.e. locally linear and lie within a structure termed the semantic tube). Due to surface statistical noise and its local nature, such trajectories for standard next token prediction are noisy and thus extend outside of the desired semantic tube. STP addresses this by constraining the trajectories during training to stay within close proximity of geodesics.

**Compliance With Llm Reviewing Policy:**

Affirmed.

**Final Justification:**

I am maintaining my original score since the strengths outweigh the somewhat limited experiments. However, small-scale experiments regarding the steerability or test-time scaling would improve the work.

**Key Questions For Authors:**

- Can this approach be used to guide test-time scaling methods like simple chain-of-thought or S1 (Muennighoff et al., 2025) to adhere to coherent trajectories and reduce hallucinations?
- See weaknesses above.

Muennighoff et al., 2025: s1: Simple test-time scaling. In arXiv

**Limitations:**

yes

**Strengths And Weaknesses:**

Strengths:
- The method is well motivated e.g. by the connection to the linear representation hypothesis (Park et al., 2024, see Figure 3) or relative geodesic representations (Yu et al., 2024).
- STP significantly improves the data efficiency of LLMs (up to 16times), which can reduce both cost and environmental impacts of training large-scale models.
- The authors perform extensive experiments covering various LLMs and model sizes.
- The paper is well-written and well structured.

Weaknesses:
- The experiments are limited to text-based datasets, extending them to multi-modal datasets and VLMs would improve the paper.
- The experiments do not cover the relation of STP and the linear representation hypothesis. Evaluating whether STP improves the “steerability” of models with linear operations like adding steering vectors (Subramani et al., 2022, Turner et al., 2023) would further improve the paper.

Subramani et al., 2022: Extracting Latent Steering Vectors from Pretrained Language Models. In ACL

Turner et al., 2023: Steering Language Models With Activation Engineering. In arXiv

---

> ### Author Rebuttal · Authors · 2026-03-31
>
> ## 1. Using STP to guide test-time scaling methods (CoT / S1)
>
> We thank the reviewer for pointing out s1 (Muennighoff et al., 2025); we will cite and discuss it in the revised Related Work section. We strongly agree that STP can broadly benefit test-time scaling methods like Chain-of-Thought (CoT). Because STP explicitly constrains hidden state trajectories to remain close to geodesics within the semantic tube, it naturally encourages more coherent generation paths, which should boost the overall consistency and quality of generated reasoning traces.
>
> Regarding s1 specifically, we see STP as a highly complementary approach. S1 achieves its results partly through a rigorous manual sample curation step to isolate a small, high-quality dataset. In contrast, our results demonstrate that STP explicitly boosts the signal-to-noise ratio during training, allowing even a randomly selected small subset of data to match the performance of the full set. Furthermore, while s1 utilizes budget-forcing to control the length of reasoning traces at test time, STP inherently improves the structural consistency of the generated trace itself. Combining STP’s representation regularization with s1’s test-time control is a highly promising direction.
>
> As a technical caveat, realizing these benefits at test-time requires the underlying model to be pretrained or fine-tuned with the STP loss, as standard pretrained models do not inherently exhibit well-behaved STP loss (as shown in Figure 4).
>
> ## 2. Extension to multi-modal datasets and VLMs
>
> We appreciate this suggestion. While extending STP to multi-modal datasets and VLMs is an exciting direction, we deliberately scoped this work to text-based LLMs. Recent works have successfully applied JEPA-like architectures to vision-language tasks (e.g., VL-JEPA [1]). However, demonstrating the efficacy of semantic tube constraints in the purely text-based LLM domain—particularly achieving up to 16x data efficiency improvements—represents a significant and necessary foundational step. We believe our extensive text-based experiments provide a solid standalone contribution, and we will explicitly note VLM extensions in our limitations/future work section.
>
> [1] VL-JEPA: Joint Embedding Predictive Architecture for Vision-language. In arXiv
>
> ## 3. Impact of STP on model "steerability"
>
> We thank the reviewer for these excellent pointers (Subramani et al., 2022; Turner et al., 2023), which we will add to our discussion. We hypothesize that STP would indeed improve the steerability of models. The linear representation hypothesis relies on the assumption that concepts can be manipulated via linear vectors. Because STP explicitly regularizes the model's hidden trajectories to be locally linear (staying within the semantic tube), an STP-trained model should yield cleaner, more robust latent steering vectors. As noted earlier, standard models do not exhibit zero STP loss (Figure 4), meaning their trajectories are noisier, which likely degrades steering effectiveness. While evaluating steering vectors empirically on an STP-trained model is beyond the current scope, the theoretical alignment is very strong, and we will add a discussion detailing how STP directly supports activation engineering.

---

> > ### Author Rebuttal · Reviewer_KkXy · 2026-04-03
> >
> > Discussing test-time scaling and the steerabilty in the related work section improves the work. However, small-scale experiments would improve the work further.

---

> > > ### Author Response · Authors · 2026-04-04
> > >
> > > Thank you for the follow-up acknowledgment. Reading your response, we realized we may have misunderstood the intent of your original question. While our initial rebuttal focused heavily on the theoretical compatibility of s1 and STP, it seems you were primarily asking for empirical evaluations involving test-time scaling and Chain-of-Thought (CoT).
> > >
> > > We would like to gently point out that these empirical results are already included in the paper. Specifically, the R1-Distill results in Figure 8(b) evaluate performance on GSM8K, a dataset that relies on CoT to answer math questions (as explained on lines 285–286, right column). Figure 8(b) demonstrates that STP significantly improves performance on this task. Furthermore, Figure 8(a) shows results for Llama 3.2 1B on GSM8K, which also yields a significant improvement over the baseline (though the margin over LLM-JEPA is smaller).
> > >
> > > We hope these existing small-scale experiments address your concern regarding empirical evidence, and we would be grateful if you took them into consideration.

---

### Decision · Program_Chairs · 2026-04-30

**Decision:**

Accept (regular)

**Comment:**

The paper proposes Semantic Tube Prediction (STP), an auxiliary cosine-similarity loss that regularizes hidden-state trajectories to lie near geodesics on a semantic manifold. Reviewers agree that the method seems simple to implement, adds negligible overhead, and improves data efficiency across multiple model families, scales, and benchmarks.

All reviewers recommend acceptance (one accept, three weak accepts), and Reviewer QBMQ raised their score from weak reject to weak accept after the rebuttal addressed concerns about overclaiming (Chinchilla scaling laws in a fine-tuning setting), single-dataset evidence for data efficiency, and theory-practice gaps.

Reviewers liked the originality of the theoretical framing (connecting the principle of least action and manifold geometry to LLM training) and the elegance of the resulting method. The experimental coverage (six datasets, six model families, three scales, five seeds with standard deviations) was noted as a strength by multiple reviewers. Reviewer QBMQ provided a particularly detailed review identifying gaps between the theoretical assumptions (infinite-width limits, zero-padded embeddings, auxiliary boundary tokens) and experimental practice. The rebuttal addressed the most critical of these and the authors agreed to reframe scaling law claims as fine-tuning-specific, provided additional data efficiency results on HellaSwag, and formalized the treatment of rotary positional embeddings within the ODE framework.

Remaining concerns include the limited evaluation to text-only structured tasks (Reviewers KkXy and yMfj) and the lack of empirical verification that the ODE approximation holds during training (Reviewer yMfj). The gaps in the theory vs practice identified by Reviewer QBMQ are only partially resolved so I encourage the authors to be explicit about which results are proven vs hypothesized in the final version.

I therefore recommend acceptance.